# AutoQRA: Joint Optimization of Mixed-Precision Quantization and Low-rank Adapters for Efficient LLM Fine-Tuning

**Changhai Zhou** [1]  **Shiyang Zhang** [3]  **Yuhua Zhou** [4]  **Qian Qiao** [5]  **Jun Gao** [4]  **Cheng Jin** [1 †]  **Kaizhou Qin** [6]
**Weizhong Zhang** [2 †]

## Abstract

Quantization followed by parameter-efficient fine-tuning has emerged as a practical paradigm for downstream adaptation under tight GPU memory constraints. However, this sequential pipeline misses the interaction between quantization bit-width and LoRA rank. Specifically, a carefully optimized quantization allocation with low calibration error does not always translate to strong fine-tuning performance, and different bit-width and rank configurations can lead to substantially different outcomes under the same memory budget. To address this limitation, we propose **AutoQRA**, a joint optimization framework that assigns both bit-width and LoRA rank for each layer during mixed-precision quantized fine-tuning. To tackle the large discrete search space and the high evaluation cost associated with repeated fine-tuning, AutoQRA decomposes the optimization process into two stages. First, it conducts a global multi-fidelity evolutionary search, where the initial population is warm-started by injecting layer-wise importance priors. This stage employs coupled operators and a performance model to efficiently screen candidate configurations. Second, trust-region Bayesian optimization is applied to locally refine promising regions of the search space and select high-utility configurations under the given memory budget. This approach enables adapter capacity to compensate for quantization noise in specific layers during training. Experiments show

that AutoQRA achieves performance close to full-precision fine-tuning with a memory footprint comparable to uniform 4-bit methods. Code is available at github.com/harrysyz99/autoqra.

## 1. Introduction

Deploying large language models (LLMs) for specific downstream tasks can be prohibitively memory intensive, which prevents many users from adapting strong base models in practice (Makridakis et al., 2023; Raiaan et al., 2024; Chang et al., 2024). A common workaround is a sequential pipeline: first quantize the pretrained backbone to fit a tight GPU memory budget, then perform parameter-efficient fine-tuning by training lightweight adapters such as LoRA while keeping the quantized backbone frozen (Hu et al., 2022; Li et al., 2023; Xu et al., 2023). In this setting, the deployment budget is a hard constraint, and the objective is the post-fine-tuning performance achieved within that budget.

Recent work has begun to exploit layer-wise heterogeneity in the quantize-then-fine-tune pipeline for LLMs. On the quantization side, mixed-precision methods allocate different bit-widths across layers according to estimated sensitivity, aiming to reduce the discrepancy between the quantized model and its full-precision counterpart (Huang et al., 2025; Lee et al., 2025). On the adaptation side, non-uniform rank allocation concentrates LoRA capacity on layers that are more important for task adaptation (Zhang et al., 2023; Zhou et al., 2025). However, we find that a bit-width allocation that appears favorable under reconstruction or calibration criteria can still lead to poor performance after fine-tuning. Moreover, under the same memory budget, different combinations of bit-width and LoRA rank can yield sharply different outcomes. The fundamental reason is that bit-width and LoRA rank interact, yet most methods treat them as independent decisions. For example, a typical performance-oriented pipeline first fixes per-layer precision using static proxies and then adjusts non-uniform ranks. This separation is misaligned with the deployment objective because the two knobs are coupled during training. Lower precision introduces quantization noise, while additional adapter capacity

---

[†]Corresponding authors. Author email: Changhai Zhou <chzhou25@m.fudan.edu.cn>. [1]College of Computer Science and Artificial Intelligence, Fudan University, Shanghai, China [2]School of Data Science, Fudan University, Shanghai, China [3]Yale University, New Haven, Connecticut, USA [4]Zhejiang University, Hangzhou, China [5]Independent Researcher [6]Obstetrics & Gynecology Hospital of Fudan University, Shanghai, China. Correspondence to: Cheng Jin <jc@fudan.edu.cn>, Weizhong Zhang <weizhongzhang@fudan.edu.cn>.

*Proceedings of the 43rd International Conference on Machine Learning*, Seoul, South Korea. PMLR 306, 2026. Copyright 2026 by the author(s).

can partially compensate for that noise through learning. Once bit-widths are fixed, the system loses the opportunity to trade redundant precision for learnability in the layers where adapters can use it most effectively, which can lead to resource misallocation.

These observations motivate a joint optimization problem that assigns a bit-width and a LoRA rank to each layer for fine-tuning. Solving this problem is difficult: the search space is large and fully discrete, making exhaustive enumeration infeasible. More importantly, low-cost proxies are unreliable because they do not model the interaction between quantization noise and adapter updates (Frantar et al., 2023; Zhao et al., 2025). Reliable evaluation therefore requires at least partial fine-tuning, which turns the search into an expensive black-box optimization problem. In many deployment settings, this search can be run offline and amortized when configurations are reused across related deployments. Nevertheless, repeated trial-and-error via fine-tuning remains prohibitively expensive.

To this end, we adopt a multi-fidelity search strategy that quickly filters poor configurations using short fine-tuning runs, and allocates longer fine-tuning runs only to a small set of promising candidates. We propose **AutoQRA**, a coarse-to-fine framework for automated quantization and rank allocation. AutoQRA uses a two-phase design that balances global coverage with local refinement.

In *Phase I*, AutoQRA performs a global multi-fidelity evolutionary search to approximate the Pareto frontier over accuracy and memory. The population is warm-started with layer-wise importance priors, and importance-guided mutations focus edits on influential layers. A learned surrogate model screens candidates and improves promotion decisions (Ru et al., 2020), while the returned frontier is formed from real measured evaluations rather than surrogate predictions. In *Phase II*, AutoQRA refines strong Phase I candidates with trust-region Bayesian optimization (Eriksson et al., 2020). We fit a Gaussian process surrogate on evaluations at the highest-fidelity setting and select configurations using Expected Improvement (EI) (Jones et al., 1998). Both phases terminate automatically when improvements saturate, using hypervolume progress for Phase I (Zitzler & Thiele, 1999) and acquisition saturation for Phase II.

Our contributions are: (1) We formulate joint per-layer bit-width and LoRA rank allocation under a strict memory budget, and explain why decoupled pipelines can be misaligned with post-fine-tuning performance. (2) We introduce AutoQRA, a two-phase coarse-to-fine framework that combines multi-fidelity evolutionary screening with trust-region Bayesian refinement to search the discrete joint space efficiently. (3) Experiments show that AutoQRA achieves performance close to full-precision fine-tuning with a memory footprint comparable to uniform 4-bit methods.

## 2. Related Work

Our work is situated at the intersection of parameter-efficient fine-tuning and automated model compression, building upon advances in quantization and black-box optimization.

### 2.1. Efficient LLM Fine-Tuning

**Quantization and Mixed-Precision.** PTQ serves as a foundation for compressing LLMs, with methods like GPTQ (Frantar et al., 2023) and AWQ (Lin et al., 2023) utilizing second-order information or activation statistics to minimize reconstruction error. This objective is well aligned with standalone quantized inference. In the setting studied here, however, the quantized weights are subsequently paired with trainable adapters, and a precision policy optimized only for frozen-backbone reconstruction need not be optimal for the final post-fine-tuning metric. To address layer-wise sensitivity, mixed-precision techniques such as SliM-LLM (Huang et al., 2025) allocate bit-widths based on Hessian spectra or salience metrics. These methods typically focus on weight precision and leave adapter capacity fixed, which is outside their original scope but leaves the bit–rank trade-off unmodeled.

**PEFT.** LoRA (Hu et al., 2022) freezes the backbone and injects trainable low-rank matrices. QLoRA (Dettmers et al., 2023) reduces memory requirements by quantizing the backbone to 4-bit, yet it retains a uniform rank assignment. Acknowledging that uniform capacity is inefficient, adaptive approaches like AdaLoRA (Zhang et al., 2023) and RankAdaptor (Zhou et al., 2025) dynamically prune or allocate ranks based on singular value importance. These methods optimize the *topology* of adapters but assume a static, uniform precision for the underlying weights, and therefore do not model the memory trade-offs available through variable quantization. Recent efforts have attempted to bridge these two paradigms. LoftQ (Li et al., 2023) and LQ-LoRA (Guo et al., 2023) propose alternating optimization schemes or initialization heuristics to align quantization with low-rank structures. While effective, they rely on localized proxies (e.g., reconstruction loss) rather than global task performance, and they typically use iterative heuristics rather than an explicit global search over the joint discrete design space.

### 2.2. Automated Search for Model Compression

Our approach also draws inspiration from Neural Architecture Search and Automated Machine Learning, which frame compression as a discrete optimization problem.

**Search Strategies for Compression.** Early works in NAS utilized Reinforcement Learning (RL) or Evolutionary Algorithms (EAs) to discover efficient architectures (Zoph & Le, 2016; Real et al., 2019). For model compression, similar

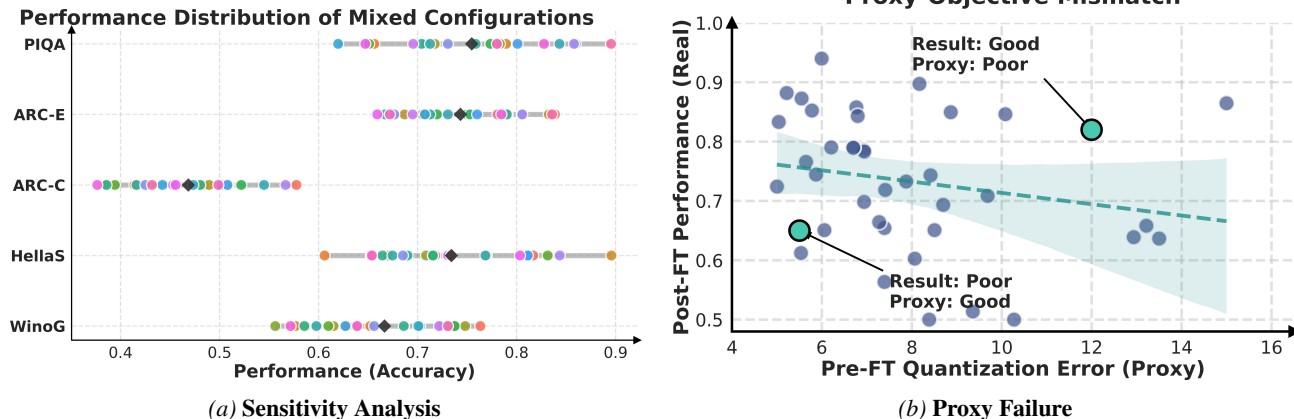

*(a)* **Sensitivity Analysis**     *(b)* **Proxy Failure**

*Figure 1.* **Empirical Motivation for Joint Optimization. (a) Impact of Joint Allocation:** We visualize the accuracy distribution of feasible mixed-precision configurations across tasks. The substantial performance spread demonstrates that distinct pairings of bit-width ($q$) and rank ($r$) can yield different outcomes even under the same memory budget. **(b) Proxy-Objective Mismatch:** Standard calibration metrics (perplexity, x-axis) correlate only moderately with post-fine-tuning accuracy (y-axis). The moderate correlation ($\rho$=0.46) and frequent rank reversals indicate that static proxies cannot reliably identify configurations where learnable adapters compensate for quantization noise.

search strategies have been applied to find per-layer quantization policies (Wang et al., 2019). However, these methods often incur prohibitive computational costs, making them impractical for LLM fine-tuning loops.

**Sample-Efficient Black-Box Optimization.** To mitigate search costs, multi-fidelity optimization has emerged as a standard. Hyperband (Li et al., 2018) and BOHB (Falkner et al., 2018) leverage cheap, low-fidelity approximations (e.g., partial epochs) to efficiently allocate resources to promising candidates. Techniques for optimizing over mixed categorical and continuous spaces, such as CoCaBO (Ru et al., 2020), demonstrate that leveraging correlations between variables can significantly accelerate convergence.

**AutoQRA.** AutoQRA builds on these search primitives but instantiates them for joint bit–rank allocation in quantized LoRA fine-tuning. The formulation enforces exact memory feasibility, couples mutations across precision and adapter rank, restricts surrogate predictions to promotion decisions, and refines measured high-fidelity candidates inside discrete trust regions. These design choices distinguish the domain-specific allocation problem studied here from general mixed-variable black-box optimization.

## 3. Methodology

We introduce **AutoQRA** (Automated Quantization–Rank Allocation), a framework designed to resolve the evaluation dilemma in joint model compression and adaptation. Auto-QRA formulates the allocation of bit-width and rank as a constrained black-box optimization problem, replacing unreliable static proxies with dynamic assessment. The search procedure uses standard optimization ingredients, including evolutionary search, Gaussian-process surrogates, and expected improvement. It adapts these ingredients to joint

allocation through exact memory feasibility, coupled bit–rank operators, measured-only Pareto selection, and discrete trust regions. To navigate the large search space efficiently, we adopt a *coarse-to-fine* strategy: Phase I approximates the global Pareto frontier via a multi-fidelity evolutionary search, while Phase II performs local Bayesian refinement to identify configurations where adapter capacity can compensate for quantization noise (Figure 2).

### 3.1. Motivation and Problem Formulation

**Motivation** We identify a dependency between quantization precision and adaptation rank. Figure 1a illustrates the performance distributions of diverse mixed bit-rank configurations across five downstream tasks. Distinct combinations of bit-width ($q$) and rank ($r$) yield substantially different outcomes even under identical memory constraints. We observe substantial performance fluctuations, with accuracy gaps exceeding 25% on tasks such as WinoGrande and ARC-Challenge. The relative performance of specific configurations, indicated by color markers, shows that downstream utility depends on the joint allocation of $q$ and $r$. Poorly matched combinations can lead to large performance drops, whereas the best observed pairings approach full-precision performance. This variation underscores that bit-width and rank cannot be optimized in isolation. Static metrics employed by decoupled methods fail to capture these non-linear interactions. We also provide a discussion on the orthogonality between backbone and adapter contributions in Appendix A.

A common strategy is to guide bit allocation using PTQ-style calibration metrics. However, once adapters are introduced, static proxies become unreliable for selecting learnable joint allocations. We randomly sample $n$=30

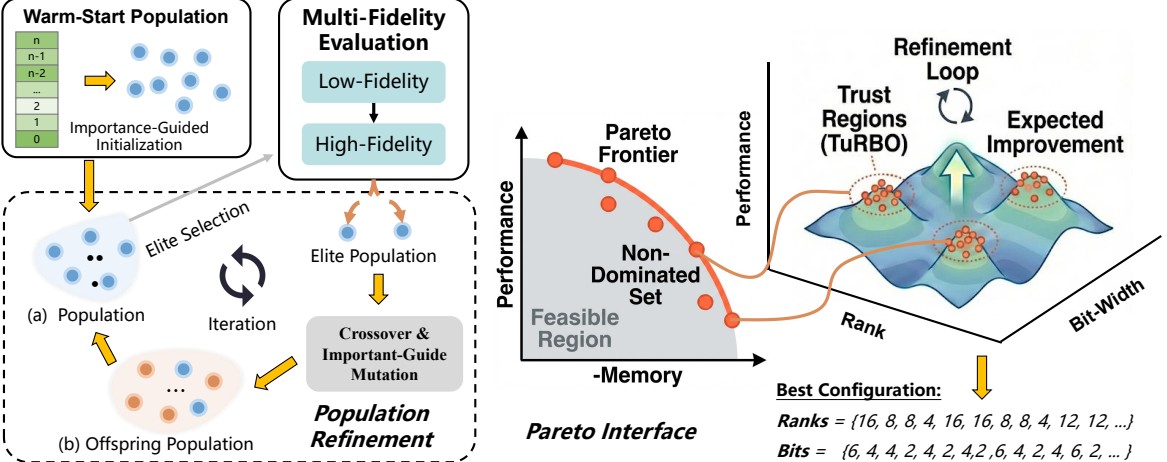

*Figure 2.* **Overview of the AutoQRA framework. Phase I** (left) approximates the global Pareto frontier via a multi-fidelity evolutionary search, utilizing importance-guided mutations and surrogate screening to navigate the discrete space. **Phase II** (right) performs a local Bayesian refinement to identify a precise operating point that maximizes user utility under the budget constraint.

feasible bit–rank configurations and compare a calibration proxy (neg. log perplexity computed with frozen weights) to the final post-fine-tuning task score. Figure 1b shows only moderate correlation and frequent rank reversals. Two failure modes are salient. First, configurations with worse proxy values can still fine-tune well when rank is assigned to learnable layers that compensate quantization noise. Second, even when proxy values are similar, post-fine-tuning accuracy can vary substantially across configurations, indicating that proxy-matched quantization error does not determine fine-tuning outcomes. Together with Figure 1a, this mismatch motivates treating joint bit–rank allocation as a constrained black-box optimization problem.

**Discussion.** A key reason for the mismatch between quantization proxies and downstream fine-tuning performance is that common proxies evaluate forward reconstruction of a frozen backbone, whereas downstream adaptation is an optimization problem carried out in a constrained update space. As a result, a bit-width allocation that appears favorable under reconstruction or calibration criteria can still lead to poor fine-tuned performance, because it may introduce error patterns that are misaligned with the correction directions expressible by low-rank updates. Equivalently, the per-layer bit pattern imposes a structural constraint on the function class of the frozen backbone, and only a subset of these constrained structures remains highly trainable under a fixed rank budget. This viewpoint is consistent with findings related to the lottery ticket hypothesis (Frankle & Carbin, 2019; Frankle et al., 2020), which suggest that under strong structural constraints only certain parameterizations or subnetworks remain highly trainable and can reach high accuracy after optimization. In our setting, different bit allocations correspond to different constrained structures, while rank allocation controls the degrees of freedom available to

compensate quantization noise during training.

**Design Space and Budget.** Consider a transformer model with $L$ layers. We define discrete search spaces for bit-widths $\mathcal{Q}$ (e.g., $\{2, 4, 8\}$) and LoRA ranks $\mathcal{R}$ (e.g., $\{4, 8, 16\}$). A configuration $C$ is a sequence of tuples specifying the precision and topology for each layer:

$$C = \{(q_\ell, r_\ell)\}_{\ell=1}^L, \quad \text{where } q_\ell \in \mathcal{Q}, \ r_\ell \in \mathcal{R}. \quad (1)$$

The total memory footprint $M(C)$ comprises the quantized backbone, trainable adapters, and quantization metadata. Let $N(W_\ell)$ be the parameter count of the dense weights at layer $\ell$. Assuming adapters utilize 16-bit precision ($p_r = 16$), the layer-wise memory cost is:

$$m_\ell(C) = \underbrace{\frac{N(W_\ell) \cdot q_\ell}{8} + m_\ell^{\text{meta}}}_{\text{Quantized Backbone}} + \underbrace{\frac{(d_{\text{in}} r_\ell + r_\ell d_{\text{out}}) \cdot p_r}{8}}_{\text{Trainable Adapters}}, \quad (2)$$

where $m_\ell^{\text{meta}}$ accounts for quantization scales and zero-points. We aim to maximize the validation performance $P(C)$ after fine-tuning, subject to a strict global memory budget $B_{\max}$.

**The Optimization Challenge.** Formally, we solve the constrained optimization problem:

$$\begin{aligned}
\underset{C}{\text{maximize}} \quad & P(C) \coloneqq \text{Metric}(\text{FineTune}(C)) \\
\text{subject to} \quad & \sum_{\ell=1}^L m_\ell(C) \le B_{\max}.
\end{aligned} \quad (3)$$

This formulation presents two challenges: (1) $P(C)$ is a *black-box* function with no closed-form gradient relative to discrete $q_\ell$ or $r_\ell$; and (2) evaluating $P(C)$ is computationally expensive, prohibiting exhaustive search over the exponential space $(|\mathcal{Q}| \times |\mathcal{R}|)^L$.

Phase I approximates the Pareto frontier $\mathcal{C}^*$ of non-dominated configurations. Phase II (Refinement) subsequently maximizes a scalarized utility function $f(C; \alpha)$ governed by a user preference parameter $\alpha \in [0, 1]$:

$$f(C; \alpha) = \alpha \cdot \hat{P}(C) - (1 - \alpha) \cdot \hat{M}(C), \qquad (4)$$

where $\hat{P}$ and $\hat{M}$ denote min-max normalized metrics.

### 3.2. Phase I: Global Multi-Fidelity Evolutionary Search

Phase I aims to construct a diverse set of *feasible* bit–rank configurations that concentrate near the performance–memory Pareto frontier, which Phase II subsequently refines. As illustrated in Figure 2, we combine (i) an evolutionary loop (Deb et al., 2002) for exploration and diversity with (ii) a Hyperband-style multi-fidelity evaluation schedule, so that evaluations at the highest-fidelity setting (largest step count) are reserved for only a small fraction of candidates.

We first define the footprint used to enforce feasibility. Following Eq. (2), the total footprint is

$$M(C) = \sum_{\ell=1}^{L} m_\ell(C). \qquad (5)$$

In all experiments, the feasibility check $M(C) \le B_{\max}$ uses the *exact* accounting from our implementation, including quantization metadata (e.g., scales and zero-points).

We use two signals to guide *joint* search: $I_q(\ell)$ for proposing bit edits and $I_r(\ell)$ for proposing rank edits. They are separated because quantization sensitivity and adaptation learnability are often mismatched, not because the optimization is decoupled. $I_q(\ell)$ measures a layer's sensitivity to low-bit perturbations on a small calibration set, while $I_r(\ell)$ measures how much update energy a layer exhibits during fine-tuning. Both signals are used only for warm start and for defining proposal distributions in our operators; they are not used as substitutes for the final fine-tuning metric. See Appendix B for $I_q(\ell)$ and $I_r(\ell)$ definitions.

We encode each configuration by an ordinal embedding $\psi(C) \in \mathbb{R}^{2L}$: each $q_\ell \in \mathcal{Q}$ and $r_\ell \in \mathcal{R}$ is mapped to its ordinal index on the corresponding ladder and then standardized, and we concatenate all layers.

**Warm start.** Initialization starts from an importance-shaped prototype and adds small perturbations for diversity:

$$q_\ell^{(0)} = \left\lfloor \tau_q(\widetilde{I}_q(\ell)) \right\rceil_\mathcal{Q}, \qquad r_\ell^{(0)} = \left\lfloor \tau_r(\widetilde{I}_r(\ell)) \right\rceil_\mathcal{R}. \qquad (6)$$

Here $\widetilde{I}_q, \widetilde{I}_r$ denote normalized scores; $\tau_q, \tau_r$ are monotone mappings from scores to discrete ladders that respect the hard constraint $M(C) \le B_{\max}$ (implemented via a greedy fill under the exact memory accounting). The initial population $\mathcal{P}_0$ is formed by this prototype plus perturbed variants, followed by feasibility repair.

**Feasibility repair.** We use a deterministic projection $\text{REPAIR}(\cdot)$ to map any infeasible candidate back into the feasible set; the operator is reused in Phase II. Let $q_\ell^-$ (resp. $r_\ell^-$) denote the nearest lower choice in $\mathcal{Q}$ (resp. $\mathcal{R}$). We define the memory saved by a single discrete downgrade as

$$\Delta M_t(\ell) = \begin{cases} M(C) - M(C \downarrow q_\ell), & t = q, \\ M(C) - M(C \downarrow r_\ell), & t = r, \end{cases} \qquad (7)$$

where $t \in \{q, r\}$, and $C \downarrow q_\ell$ replaces $q_\ell$ by $q_\ell^-$ (analogously for $r_\ell$). While $M(C) > B_{\max}$, we apply the downgrade that minimizes sensitivity per saved memory:

$$(\ell^\star, t^\star) = \arg \min_{\ell, \, t \in \{q, r\}} \frac{\widetilde{I}_t(\ell) + \epsilon}{\Delta M_t(\ell)} \quad \text{s.t.} \quad \Delta M_t(\ell) > 0, \qquad (8)$$

with $I_q$ used for $t = q$ and $I_r$ used for $t = r$. This rule removes capacity from low-sensitivity layers and from the knob (bit or rank) that yields the largest memory relief for the smallest expected damage, terminating once feasibility is restored.

**Variation operators.** Offspring are generated by two complementary mutations, each followed by $\text{REPAIR}(\cdot)$. A sensitivity-guided mutation selects a layer according to

$$\Pr_t(\ell) = \frac{I_t(\ell)^\gamma}{\sum_{j=1}^{L} I_t(j)^\gamma}, \qquad t \in \{q, r\}, \qquad (9)$$
$$(t = q: \text{bit updates}; \ t = r: \text{rank updates}) \, .$$

where $\gamma > 0$ controls the concentration of importance sampling. It then changes $q_\ell$ or $r_\ell$ by one adjacent discrete step. A memory-balanced coupled mutation applies a primary memory-increasing edit (e.g., $q_\ell \uparrow$ or $r_\ell \uparrow$), then performs compensating memory-decreasing edits (possibly on different layers) until feasibility is restored. This global compensation avoids the scale mismatch of within-layer "iso-memory" pairing and keeps the search concentrated near the constraint boundary.

**Multi-fidelity evaluation.** We parameterize evaluation cost by the number of fine-tuning steps $T$ (equivalently, a fractional number of epochs) and use an increasing ladder $0 < T_1 < \cdots < T_S$ with pruning factor $\eta > 1$. Evaluations at smaller $T_s$ provide inexpensive but noisier estimates $P(C; T_s)$ (low-fidelity, LF), while $T_S$ denotes the *largest step count used in our search* (high-fidelity, HF) and is not assumed to correspond to fully converged fine-tuning. When a candidate is promoted from $T_s$ to $T_{s+1}$, it *continues training from its checkpoint at $T_s$* rather than restarting, so that $P(C; T_{s+1})$ is directly comparable across candidates. To reduce variance in LF ranking, we fix the optimizer hyperparameters, random seed, and data order across candidates.

**Surrogate screening.** Multi-fidelity screening is used exclusively for promotion decisions. At stage $s$, we train $\Phi_s$ to predict the HF score $P(C; T_S)$ from LF observations.

Let $\mathcal{D}_s = \{(C_i, P(C_i; T_s), P(C_i; T_S))\}$ be the paired set collected so far (across generations). With feature vector $x_i = [P(C_i; T_s), \log M(C_i), \psi(C_i)]$, we fit $\Phi_s(\cdot; \theta_s)$ by regression:

$$\theta_s^\star = \arg\min_\theta \sum_{(C_i, \cdot) \in \mathcal{D}_s} \rho(\Phi_s(x_i; \theta) - P(C_i; T_S)) + \lambda \|\theta\|_2^2, \tag{10}$$

where $\rho(\cdot)$ is the Huber loss to mitigate occasional LF outliers. When $|\mathcal{D}_s|$ is insufficient early on, promotion falls back to ranking by the measured $P(C; T_s)$. Crucially, Pareto selection uses *measured* $P(C; T_S)$ rather than surrogate predictions, ensuring the returned frontier is supported by evaluations at $T_S$.

**Update and termination.** After each generation, we update the population using NSGA-II with constrained domination: feasible candidates dominate infeasible ones; among infeasible candidates, lower constraint violation is preferred. The loop terminates when the dominated hypervolume (Appendix C.2) of the feasible front stabilizes (relative improvement below $\epsilon_{\text{hv}}$ for $\Delta$ generations), yielding a compact feasible Pareto set for Phase II.

### 3.3. Phase II: Local Bayesian Refinement

Phase I is primarily responsible for *coverage*: it explores the combinatorial bit–rank space under the hard constraint $M(C) \leq B_{\max}$ and returns a diverse set of *measured* candidates evaluated at $T_S$ near the Pareto frontier. Phase II is responsible for *selection and sharpening*: given a user preference $\alpha$ (Eq. (4)), we identify a single operating point by improving the scalarized utility $f(C; \alpha)$ using a small number of additional evaluations at $T_S$.

A natural concern is that a handful of Phase II evaluations may be insufficient in the original exponential space. We therefore do *not* optimize over the full space in Phase II. Instead, Phase II operates on a reduced search region that Phase I has already verified to be feasible and competitive at $T_S$. Moreover, rather than committing to a single local basin, we follow the multi-region trust-region principle of TuRBO (Eriksson et al., 2020): we maintain several discrete trust regions around multiple strong Phase I solutions and allocate the limited Phase II evaluations to the region with the highest predicted improvement.

Let $\mathcal{D}^{\text{hi}} = \{(C_i, y_i)\}_{i=1}^{n_0}$ collect all *measured* evaluations at $T_S$ after Phase I. Each $C_i$ is evaluated at $T_S$ and satisfies $M(C_i) \leq B_{\max}$.

For each entry we record

$$\begin{aligned} y_i &\coloneqq f(C_i; \alpha) \coloneqq \alpha \hat{P}(C_i) - (1-\alpha) \hat{M}(C_i), \\ P(C_i) &\coloneqq P(C_i; T_S) \end{aligned} \tag{11}$$

We set $y_0^+ = \max_{1 \leq i \leq n_0} y_i$ and initialize $J$ trust-region centers by taking the top candidates in $\mathcal{D}^{\text{hi}}$ while en-

forcing diversity in atomic distance: we greedily select $C_0^{(1)}, \ldots, C_0^{(J)}$ such that each new center satisfies $d_{\text{atom}}(C_0^{(j)}, C_0^{(j')}) \geq \Delta_{\text{div}}$ for all $j' < j$. The global incumbent is the best measured one in $\mathcal{D}^{\text{hi}}$ (ties broken by smaller $M(C_i)$).

We model the utility landscape with a Gaussian process surrogate in the standardized ordinal embedding $\psi(C) \in \mathbb{R}^{2L}$ used throughout. We place a GP prior on a latent function $g(C)$ and assume noisy observations

$$g(C) \sim \mathcal{GP}(0, k(C, C')), \ y = g(C) + \varepsilon, \ \varepsilon \sim \mathcal{N}(0, \sigma_n^2), \tag{12}$$

refitting hyperparameters by maximizing the GP marginal likelihood on the updated $\mathcal{D}^{\text{hi}}$. We use a Matérn-5/2 kernel in the embedded space:

$$\begin{aligned} k(C, C') &= \sigma_f^2 \left(1 + \sqrt{5}\, r + \frac{5}{3} r^2\right) \exp(-\sqrt{5}\, r), \\ r &= \|\psi(C) - \psi(C')\|_2 / \ell_{\text{gp}}. \end{aligned} \tag{13}$$

Phase II proposes candidates only within discrete trust regions. An *atomic edit* changes exactly one variable to an adjacent value on its ladder (either a neighbor in $\mathcal{Q}$ for $q_\ell$ or a neighbor in $\mathcal{R}$ for $r_\ell$). Let $d_{\text{atom}}(C, C')$ be the minimum number of atomic edits needed to transform $C$ into $C'$. For region $j$ at iteration $t$, with center $C_t^{(j)}$ and integer radius $\delta_{j,t}$, we first define the pre-repair neighborhood

$$\mathcal{B}_{j,t} \triangleq \{ C' \mid d_{\text{atom}}(C', C_t^{(j)}) \leq \delta_{j,t} \}, \tag{14}$$

and then project every candidate to feasibility using the same operator $\text{REPAIR}(\cdot)$ as in Phase I:

$$\Omega_{j,t} \triangleq \{ \text{REPAIR}(C') \mid C' \in \mathcal{B}_{j,t} \}. \tag{15}$$

Thus every $C \in \Omega_{j,t}$ satisfies $M(C) \leq B_{\max}$ under the same exact accounting as Phase I. We then form the union $\Omega_t = \bigcup_{j=1}^{J} \Omega_{j,t}$. When $\Omega_t$ is too large to enumerate, we subsample a fixed-size pool from each neighborhood before applying $\text{REPAIR}(\cdot)$, using the same sensitivity-guided proposal distribution as in Phase I.

Given the GP posterior mean $\mu_t(C)$ and standard deviation $\sigma_t(C)$, we use Expected Improvement (EI) with respect to the best measured utility $y_t^+ = \max\{y_i : (C_i, y_i) \in \mathcal{D}^{\text{hi}}\}$:

$$\begin{aligned} z_t(C) &\coloneqq \frac{\mu_t(C) - y_t^+}{\sigma_t(C)}, \\ \text{EI}_t(C) &\coloneqq (\mu_t(C) - y_t^+)\, \Phi(z_t(C)) + \sigma_t(C)\, \phi(z_t(C)), \end{aligned} \tag{16}$$

We select the next configuration by scanning the pool $\Omega_t$:

$$C_{t+1} = \arg\max_{C \in \Omega_t} \text{EI}_t(C), \tag{17}$$

evaluate it at $T_S$ to obtain $y_{t+1} = f(C_{t+1}; \alpha)$, and update $\mathcal{D}^{\text{hi}} \leftarrow \mathcal{D}^{\text{hi}} \cup \{(C_{t+1}, y_{t+1})\}$. Surrogate predictions are

*Table 1.* **Main results across four backbones.** We report task-average accuracy (%), average weight precision (AvgBit), average LoRA rank (AvgRank), and total memory footprint (Mem, GB). Bold / underline indicate best / second best per backbone.

| Method | LLaMA3.1-8B | | | | LLaMA3.2-3B | | | | Qwen2.5-7B | | | | Qwen2.5-3B | | | |
|---|---|---|---|---|---|---|---|---|---|---|---|---|---|---|---|---|
| | Avg↑ | AvgBit↓ | AvgRank | Mem↓ | Avg↑ | AvgBit↓ | AvgRank | Mem↓ | Avg↑ | AvgBit↓ | AvgRank | Mem↓ | Avg↑ | AvgBit↓ | AvgRank | Mem↓ |
| LoRA (FP16) | 69.94 | 16.00 | 16.00 | 20.50 | 65.40 | 16.00 | 16.00 | 10.40 | 71.33 | 16.00 | 16.00 | 20.20 | 65.53 | 16.00 | 16.00 | 10.84 |
| QLoRA (4-bit) | 67.45 | 4.00 | 16.00 | 15.22 | 64.43 | 4.00 | 16.00 | 8.90 | 69.01 | 4.00 | 16.00 | 15.60 | 62.89 | 4.00 | 16.00 | 8.16 |
| AdaLoRA (4-bit) | 66.36 | 4.00 | 15.84 | 14.92 | 61.88 | 4.00 | 15.73 | 8.60 | 69.05 | 4.00 | 15.65 | 15.40 | 62.51 | 4.00 | 15.92 | 8.70 |
| LoftQ (4-bit) | 68.65 | 4.00 | 16.00 | 15.13 | 64.91 | 4.00 | 16.00 | 8.83 | 69.35 | 4.00 | 16.00 | 15.33 | 62.71 | 4.00 | 16.00 | 8.21 |
| LQ-LoRA | 67.82 | 3.78 | 16.00 | 20.12 | 63.63 | 3.85 | 16.00 | 7.12 | 66.91 | 3.85 | 16.00 | 18.52 | 62.86 | **3.63** | 16.00 | 7.05 |
| AMQ+LoRA | 67.75 | 3.88 | 16.00 | 14.45 | 63.31 | 3.91 | 16.00 | 7.22 | 70.86 | 3.91 | 16.00 | 14.90 | 63.30 | 4.00 | 16.00 | 8.17 |
| AMQ+AdaLoRA | 67.63 | 3.93 | **10.18** | 14.40 | 63.38 | 3.96 | **9.68** | 7.25 | 70.66 | 3.96 | **10.18** | 14.70 | 64.88 | 3.80 | 15.84 | 8.65 |
| AutoQRA (≤4-bit) | 69.83 | **3.75** | 10.50 | **13.08** | 65.58 | **3.64** | 11.14 | **6.72** | 71.35 | **3.71** | 10.57 | **11.95** | 66.33 | 3.72 | **9.78** | **6.45** |
| AutoQRA (Opt) | **70.45** | 5.25 | 12.25 | 17.32 | **66.16** | 5.14 | 12.57 | 8.41 | **73.19** | 5.36 | 12.70 | 17.24 | **68.05** | 5.22 | 12.00 | 8.31 |

used only for proposing evaluations; the final returned configuration is always supported by measured utility.

We adapt the trust-region radius only for the region that generated $C_{t+1}$. Let $j(t)$ denote the selected region (i.e., $C_{t+1} \in \Omega_{j(t),t}$). For every region $j \in \{1, \ldots, J\}$,

$$\delta_{j,t+1} = \begin{cases} \min\{\kappa_\uparrow \delta_{j,t}, \delta_{\max}\}, & j = j(t), \ y_{t+1} > y_t^+, \\ \max\{\kappa_\downarrow \delta_{j,t}, \delta_{\min}\}, & j = j(t), \ y_{t+1} \leq y_t^+, \\ \delta_{j,t}, & j \neq j(t), \end{cases} \tag{18}$$

and we update the corresponding center by

$$C_{t+1}^{(j)} = \begin{cases} C_{t+1}, & j = j(t), \ y_{t+1} > y_t^+, \\ C_t^{(j)}, & \text{otherwise.} \end{cases} \tag{19}$$

The loop terminates when either (i) no meaningful improvement is predicted,

$$\max_{C \in \Omega_t} \text{EI}_t(C) < \epsilon_{\text{ei}}, \tag{20}$$

or (ii) a hard cap of $N_{\max}$ Phase II iterations is reached. We return the best measured feasible configuration $C^\star = \arg\max_{(C_i, y_i) \in \mathcal{D}^{\text{hi}}} y_i$.

# 4. Experiments

## 4.1. Experimental Setup

**Datasets and LLMs.** We fine-tune on Alpaca52k and HC3 (Taori et al., 2023), and evaluate zero-/few-shot on BoolQ (Clark et al., 2019), PIQA (Bisk et al., 2020), HellaSwag (Zellers et al., 2019), WinoGrande (Sakaguchi et al., 2021), ARC-E/ARC-C (Clark et al., 2018), OpenBookQA (Mihaylov et al., 2018), and MMLU (Hendrycks et al., 2021). Backbones include LLaMA-3.1/3.2 (Grattafiori et al., 2024) and Qwen-2.5 (Qwen et al., 2025).

**Baselines.** We compare with LoRA (FP16) (Hu et al., 2022), QLoRA (4-bit) (Dettmers et al., 2023), AdaLoRA (Zhang et al., 2023), LoftQ (Li et al., 2023), and LQ-LoRA (Guo et al., 2024). To isolate the value of *joint* bit–rank allocation, we further include a decoupled baseline that first runs

an automatic mixed-precision quantization allocator AMQ (Lee et al., 2025) and then fine-tunes with either LoRA or AdaLoRA(**AMQ+L, AMQ+AL**).

**Implementation Details.** All methods are implemented in PyTorch using the Transformers/PEFT/BitsAndBytes stack and run on NVIDIA A100 GPUs. All methods use the same backbone, datasets, and training protocol; they differ only in how they allocate per-layer $(q_\ell, r_\ell)$ under the same global memory budget $B_{\max}$. We describe the full AutoQRA search schedule (Phase I/II step ladder, low- vs. high-fidelity evaluations, and early stopping) in Appendix C.

## 4.2. Main Results

Table 1 reports the main results on four backbones. We summarize each method by (i) task-average accuracy and (ii) the resource triple (AvgBit, AvgRank, Mem) computed under the same memory accounting used by our implementation. Overall, AutoQRA achieves a strong accuracy–memory trade-off: under the AvgBit≤ 4 regime, AutoQRA consistently improves over uniform 4-bit baselines (QLoRA/AdaLoRA/LoftQ) while using lower effective precision and a smaller footprint; when allowing mixed precision under the same protocol, AutoQRA (Opt) slightly exceeds FP16 LoRA in the reported task average on all four backbones with substantially reduced weight precision. In particular, AutoQRA (AvgBit≤ 4) is the best-performing ≤4-bit method across all backbones, while reducing footprint by 12–22% relative to uniform 4-bit baselines. It also achieves these gains with markedly smaller AvgRank, highlighting the benefit of coordinated bit–rank allocation under fixed memory. Decoupled pipelines remain less competitive under the same memory budget $B_{\max}$. AMQ+LoRA/AdaLoRA first allocates bits by a static quantization objective and only then tunes ranks, which does not explicitly exploit the compensatory interplay between quantization noise and adapter capacity. Per-task accuracies are reported in Appendix D.

**Evidence of bit-width and rank compensation.** Figure 3 provides direct evidence for the compensation effect that

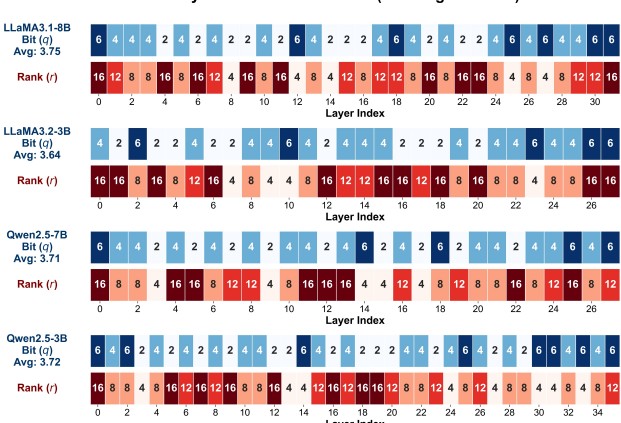

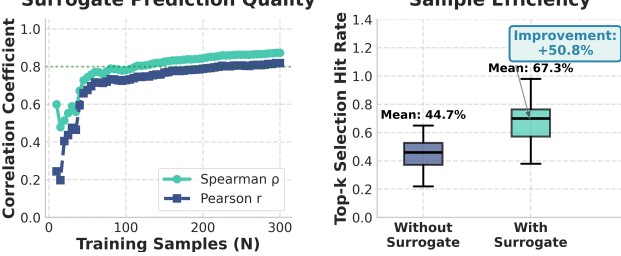

Figure 3. **Layer wise configurations found by AutoQRA show a compensation pattern.** Layers assigned lower bit-widths are often paired with higher ranks, suggesting that adapter capacity is

Figure 4. **Surrogate Quality.** Surrogate accuracy improves with paired data and boosts top-3 promotion hit rate.

motivates our joint search. Across models, AutoQRA assigns high ranks to many of the layers that are quantized more aggressively, while keeping ranks small in layers that retain higher precision. This produces a consistent negative association between $q_\ell$ and $r_\ell$ across layers, despite similar average bit-widths across the compared models. These patterns indicate that better fine-tuning performance under a fixed memory budget is achieved through coordinated allocation of bit-widths and ranks, rather than by optimizing quantization error alone.

### 4.3. Screening surrogate quality.

We evaluate whether low-fidelity (LF) observations can be reliably mapped to performance at the highest-fidelity setting $T_S$ by our screening surrogate. Figure 4 (left) reports predictive quality as the amount of paired LF and $T_S$ data increases: both Spearman and Pearson correlations rise quickly and then stabilize. Figure 4 (right) summarizes how surrogate screening affects promotion decisions. Each boxplot aggregates the *top-3 selection hit rate* over repeated trials, where the hit rate measures how often the three candidates selected for promotion align with the oracle top performers judged at $T_S$. Surrogate-guided promotion shifts the entire distribution upward and increases the mean hit rate from 44.7% (ranking by LF scores alone) to 67.3%, corresponding to a 50.8% relative improvement.

*Table 2.* **Search and per-step overhead on Qwen2.5-3B.**

| Method | Search | Time/step (s) |
|---|---|---|
| QLoRA (4-bit) | 0 min | 2.41 |
| LoftQ (4-bit) | $\sim$10 min | 2.42 |
| AdaLoRA (4-bit) | 0 min | 2.81 |
| AMQ+LoRA | >3.6 h | 1.91 |
| AutoQRA | 55 min | 1.92 |

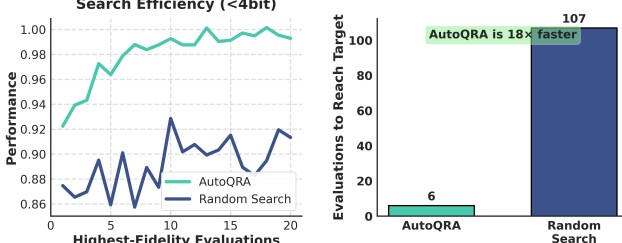

Figure 5. **Search efficiency analysis.** (Left) Best validation performance versus the number of evaluations at the largest search budget $b_S$. AutoQRA improves rapidly and consistently performs better than random search. (Right) Number of largest-budget evaluations required to reach a target accuracy. AutoQRA needs 6 evaluations compared to 107 for random search, yielding an $18\times$ reduction in expensive evaluations.

### 4.4. Search efficiency and sample complexity.

To validate the effectiveness of our optimization strategy, we benchmark AutoQRA against random search under the same low-bit search setting. Figure 5 (left) reports the best validation performance found as a function of the number of HF evaluations at $T_S$. AutoQRA improves rapidly, identifying strong configurations within the first few trials and then maintaining a stable trajectory. In contrast, random search shows large variance and struggles to locate high-performing regions in the combinatorial space. We further quantify this advantage in Figure 5 (right) by measuring the number of HF evaluations at $T_S$ required to reach a fixed performance target. Random search requires 107 evaluations to reach the target, whereas AutoQRA reaches the same target with 6 evaluations. We measure wall-clock overhead on Qwen2.5-3B. AutoQRA completes the joint bit–rank search in 55 minutes, compared with more than 3.6 hours for AMQ, which searches bit-width only (Table 2). The selected AutoQRA configuration trains at 1.92 seconds per step, close to AMQ+LoRA (1.91 seconds) and faster than AdaLoRA (2.81 seconds). AdaLoRA performs dynamic SVD-based rank updates during training, while QLoRA and LoftQ use a different quantization backend from the HQQ mixed-precision path used by AutoQRA and AMQ.

### 4.5. Robustness and generalization.

We evaluate AutoQRA under tighter memory budgets, transfer of searched configurations across datasets, and generalization to code generation and larger backbones. Table 3 summarizes these results on Qwen2.5-3B unless otherwise noted. Reducing the memory budget to $0.7\times$ the default

*Table 3.* **Robustness and generalization summary.**

| Setting | Comparison | Score |
|---------|-----------|-------|
| Budget | $1.0\times$ / $0.7\times$ combined Avg | 64.55 / 63.92 |
| Budget | $1.0\times$ / $0.7\times$ HumanEval | 41.96 / 41.39 |
| Transfer | HC3 search / Alpaca transfer | 65.93 / 65.54 |
| Code | LoRA / AutoQRA (Opt), HumanEval | 43.19 / 43.07 |
| 14B | QLoRA / AutoQRA ($\leq$4-bit) Avg | 79.69 / 80.30 |
| 14B | LoRA / AutoQRA (Opt) Avg | 80.31 / 80.71 |

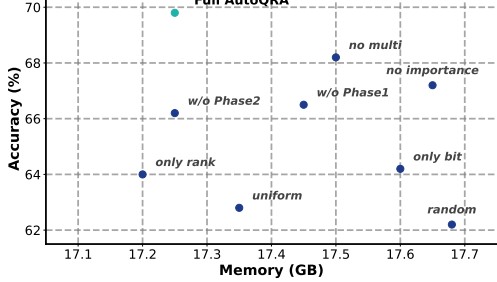

*Figure 6.* **Ablation.** Accuracy vs. memory under the same $B_{\max}$; the full method (green) dominates ablated variants (blue).

$\leq$4-bit budget changes the combined average over the five commonsense tasks plus HumanEval from 64.55 to 63.92, while HumanEval pass@1 changes from 41.96 to 41.39. For transfer, a configuration searched on Alpaca obtains 65.54 on HC3, compared with 65.93 for a configuration searched directly on HC3 and 62.45 for QLoRA in Appendix F. On HumanEval, AutoQRA (Opt) reaches 43.07 pass@1 compared with 43.19 for FP16 LoRA. On Qwen2.5-14B, AutoQRA ($\leq$4-bit) improves over QLoRA while using lower average precision and rank, and AutoQRA (Opt) obtains the highest average score among the compared methods.

### 4.6. Ablation Study

We ablate key components of AutoQRA under a fixed memory budget: (i) removing the warm-start / importance prior, (ii) disabling Phase I global search (BO only), (iii) disabling Phase II local refinement (EA only), (iv) optimizing only one axis (bit-only or rank-only), and (v) removing multi-fidelity and/or surrogate screening. Across tasks, the full system consistently yields the best accuracy–memory trade-off, while removing global exploration or feasibility-aware search degrades the frontier most strongly.

We find that without the feasibility projection REPAIR($\cdot$), a large fraction of generated candidates violate the hard memory budget, reducing effective search throughput and weakening Pareto coverage. Detailed diagnostics for repair behavior and a sensitivity study of key search hyperparameters are provided in Appendix E.

## 5. Conclusion

We introduced AutoQRA, a framework for jointly allocating quantization bit-width and LoRA rank at the layer level. By searching over these two coupled choices together, Auto-

QRA addresses a limitation of sequential quantize-then-adapt pipelines, which cannot explicitly trade precision against adapter capacity during fine-tuning. The proposed coarse-to-fine search combines multi-fidelity evolutionary screening with trust-region Bayesian refinement, allowing it to explore the discrete design space with a limited number of high-fidelity evaluations. Across LLaMA and Qwen backbones, AutoQRA improves over uniform 4-bit baselines and approaches full-precision LoRA performance while using lower average precision. The learned configurations also reveal a consistent compensation pattern: layers assigned lower precision often receive higher LoRA ranks. These results suggest that coordinated bit–rank allocation is an effective way to use a fixed memory budget for quantized fine-tuning.

**Limitations and future work.** AutoQRA uses fixed default search hyperparameters together with early stopping; it does not automatically allocate search resources as a function of model scale or task heterogeneity. The sensitivity study indicates that smaller backbones and mixed-task evaluations benefit from a wider Phase I promotion set and a few more Phase II iterations, whereas larger backbones are well served by the default budget. A scale- and task-aware controller would require an additional meta-optimization layer and broader validation, which we leave to future work.

## Impact Statement

This paper presents work that advances efficient LLM fine-tuning. Our primary goal is to make strong model adaptation more accessible under limited GPU memory. By reducing the memory footprint required for fine-tuning, AutoQRA can lower the computational barrier for researchers and developers with constrained hardware resources. The method may also reduce resource use when a lower-memory adaptation pipeline replaces a larger full-precision setup. We do not foresee specific negative societal consequences beyond the general risks associated with model compression and large language models.

## Acknowledgements

We thank our faculty mentors and colleagues for guidance, feedback, and helpful discussions. This work was supported by the National Natural Science Foundation of China (Grant No. 6247 n2097), the Shanghai Municipal Science and Technology Commission (Grant No. 24511106102), the AI for Science Foundation of Fudan University (FudanX24AI028), and the Fudan Kunpeng & Ascend Center of Cultivation. The computations in this research were performed on the CFFF platform of Fudan University.

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

# Appendix

## A. Orthogonal Sensitivity and Compensatory Potential

While layer-wise heterogeneity is well-documented, the primary rationale for joint optimization lies in the *orthogonality* of sensitivity profiles. As illustrated in Figure 7, layers highly sensitive to quantization (requiring high bit-width) often differ from those requiring high adaptation capacity (requiring high rank). Conventional sequential pipelines overlook this nuance: a quantization-first approach might conservatively assign 4-bit precision to a sensitive layer, ignoring that a high-rank adapter could effectively compensate for the noise induced by aggressive 2-bit quantization. Conversely, independent optimization might allocate high ranks to layers robust to quantization but contributing little to task adaptation. Therefore, effective allocation requires capturing this *compensatory interplay*: trading precision for learnability based on layer-specific joint sensitivity.

## B. Additional Details for Phase I

**Exact memory decomposition.** For feasibility checks we use the exact memory accounting implemented in our code, including quantization metadata such as scales and zero-points. For readability, the footprint can be decomposed as

$$M(C) = \sum_{\ell=1}^{L} \left( m_\ell^W(q_\ell) + m_\ell^A(r_\ell) + m_\ell^{\text{meta}}(q_\ell) \right), \quad (21)$$

where $m_\ell^W$ is the quantized backbone storage, $m_\ell^A$ is the adapter storage, and $m_\ell^{\text{meta}}$ captures quantization metadata. Letting $N(W_\ell)$ denote the number of backbone parameters at layer $\ell$, we write

$$m_\ell^W(q_\ell) = \frac{N(W_\ell)\, q_\ell}{8}, \qquad m_\ell^A(r_\ell) = \frac{N(A_\ell, r_\ell)\, p_r}{8}, \quad (22)$$

with $p_r = 16$ for FP16 adapters. For LoRA applied to a set of linear maps $\mathcal{S}_\ell$ in layer $\ell$, the adapter parameter count is

$$N(A_\ell, r_\ell) = \sum_{W \in \mathcal{S}_\ell} r_\ell \big( d_{\text{in}}(W) + d_{\text{out}}(W) \big), \quad (23)$$

following the standard LoRA parameterization (Hu et al., 2022).

**Importance signals for warm start and proposals.** We define two layer-wise signals used only for warm start and proposal distributions in Phase I (and for feasibility repair), not as substitutes for the final fine-tuning metric. For bit-width proposals, we use a gradient-weighted quantization residual on a small calibration set. Let $\mathcal{S}_\ell^W$ denote backbone

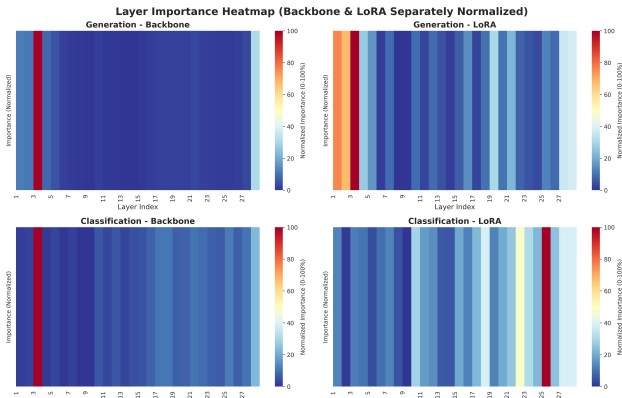

*Figure 7.* **Orthogonal sensitivity profiles in Qwen3-1.7B.** Normalized importance scores for quantization (backbone weight sensitivity, left) and adaptation (LoRA rank sensitivity, right). These distributions diverge: layers requiring high precision to suppress quantization noise do not necessarily demand high-rank adapters for downstream tasks. This *sensitivity mismatch* motivates joint allocation, as independent optimization does not capture these compensatory trade-offs.

matrices in layer $\ell$, let $q_{\min} = \min \mathcal{Q}$, and let $Q_{q_{\min}}(\cdot)$ denote quantization at $q_{\min}$. With a diagonal Fisher proxy $\widehat{F}_W \approx \mathbb{E}\left[ \left( \frac{\partial \mathcal{L}}{\partial W} \right)^{\odot 2} \right]$, we define

$$I_q(\ell) = \sum_{W \in \mathcal{S}_\ell^W} \left\langle \widehat{F}_W, \, \left( W - Q_{q_{\min}}(W) \right)^{\odot 2} \right\rangle. \quad (24)$$

For rank proposals, we run a short probe fine-tuning for $K$ steps and collect the averaged gradient matrix for each LoRA-targeted projection $W \in \mathcal{S}_\ell$:

$$\overline{G}_W = \frac{1}{K} \sum_{k=1}^{K} \frac{\partial \mathcal{L}_k}{\partial W}. \quad (25)$$

Let $\{\sigma_j(\overline{G}_W)\}$ be the singular values of $\overline{G}_W$. We define the rank signal by the leading spectral energy:

$$I_r(\ell) = \sum_{W \in \mathcal{S}_\ell} \sum_{j=1}^{J_0} \sigma_j(\overline{G}_W)^2, \quad (26)$$

where $J_0$ is a small cutoff (e.g., $J_0 = 8$) to emphasize dominant update directions. In Phase I we min–max normalize these scores across layers and use them for warm-start initialization, importance-guided proposals, and feasibility repair.

## C. Experimental Details

### C.1. Phase I multi-fidelity step allocation

We use a Hyperband/SHA-style ladder with step counts $0 < T_1 < \cdots < T_S$ and pruning factor $\eta$. In our default

*Table 4.* **Detailed Task-wise accuracy.** Bold indicates best result per backbone.

| Model | Method | Bit | Rank | Mem | Avg | ARC-C | ARC-E | BoolQ | GSM8K | HellaS | OBQA | PIQA | WinoG |
|---|---|---|---|---|---|---|---|---|---|---|---|---|---|
| LLaMA-3.1-8B | LoRA | 16.00 | 16.00 | 20.50 | 69.94 | 56.14 | 83.88 | 83.37 | **54.06** | 79.44 | **45.60** | 82.10 | **74.90** |
| | QLoRA | 4.00 | 16.00 | 15.22 | 67.45 | 54.15 | 82.20 | 81.95 | 44.15 | 78.50 | 43.60 | 81.40 | 73.64 |
| | AdaLoRA | 4.00 | 15.84 | 14.92 | 66.36 | 52.40 | 81.65 | 81.50 | 42.80 | 78.10 | 40.80 | 81.10 | 72.55 |
| | LoftQ | 4.00 | 16.00 | 15.13 | 68.65 | 54.80 | 82.65 | 82.15 | 51.10 | 78.65 | 45.00 | 81.45 | 73.40 |
| | LQ-LoRA | 3.78 | 16.00 | 20.12 | 67.82 | 54.75 | 82.45 | 82.25 | 44.95 | 79.10 | 44.20 | 81.60 | 73.25 |
| | AMQ+L | 3.88 | 16.00 | 14.45 | 67.75 | 54.55 | 82.60 | 82.40 | 44.50 | 78.90 | 44.00 | 81.42 | 73.61 |
| | AMQ+AL | 3.93 | 10.18 | 14.40 | 67.63 | 54.45 | 82.35 | 82.15 | 44.25 | 78.60 | 43.80 | 81.76 | 73.69 |
| | **AutoQRA ($\leq$4-bit)** | 3.75 | 10.50 | 13.08 | 69.83 | 56.12 | 84.20 | 83.35 | 53.40 | 79.50 | 45.60 | 82.40 | 74.10 |
| | **AutoQRA (Opt)** | 5.25 | 12.25 | 17.32 | **70.45** | 57.85 | 84.90 | 83.86 | 53.90 | **79.95** | 45.20 | 83.10 | 74.80 |
| LLaMA-3.2-3B | LoRA | 16.00 | 16.00 | 10.40 | 65.40 | 48.38 | **79.78** | 81.10 | 47.31 | 75.75 | 42.20 | 78.62 | 70.09 |
| | QLoRA | 4.00 | 16.00 | 8.90 | 64.43 | 47.18 | 77.53 | 77.61 | 46.22 | 74.79 | 42.40 | 78.73 | 70.96 |
| | AdaLoRA | 4.00 | 15.73 | 8.60 | 61.88 | 46.42 | 74.79 | 74.68 | 36.50 | 73.92 | 41.20 | 77.97 | 69.53 |
| | LoftQ | 4.00 | 16.00 | 8.83 | 64.91 | 47.80 | 77.90 | 77.95 | 47.50 | 75.10 | 42.80 | 79.00 | 71.20 |
| | LQ-LoRA | 3.85 | 16.00 | 7.12 | 63.63 | **51.45** | 79.20 | 79.05 | 36.40 | 75.25 | 40.40 | 78.20 | 69.10 |
| | AMQ+L | 3.91 | 16.00 | 7.22 | 63.31 | 46.90 | 75.90 | 76.80 | 42.10 | 73.90 | 41.50 | 78.90 | 70.50 |
| | AMQ+AL | 3.96 | 9.68 | 7.25 | 63.38 | 47.30 | 79.25 | 79.40 | 36.85 | 75.50 | 40.80 | 78.50 | 69.45 |
| | **AutoQRA ($\leq$4-bit)** | 3.64 | 11.14 | 6.72 | 65.58 | 47.97 | 78.96 | 78.59 | 48.93 | 76.97 | 42.80 | 79.15 | 71.20 |
| | **AutoQRA (Opt)** | 5.14 | 12.57 | 8.41 | **66.16** | 48.00 | 79.60 | 79.70 | **49.72** | 77.29 | **43.80** | 79.40 | 71.80 |
| Qwen-2.5-7B | LoRA | 16.00 | 16.00 | 20.20 | 71.33 | 53.16 | 79.21 | 84.43 | 75.24 | 77.87 | 46.00 | 81.66 | 73.04 |
| | QLoRA | 4.00 | 16.00 | 15.60 | 69.01 | 54.27 | 79.04 | 83.73 | 73.50 | 78.04 | 43.80 | 76.39 | 63.30 |
| | AdaLoRA | 4.00 | 15.65 | 15.40 | 69.05 | 50.85 | 80.40 | 80.00 | 69.80 | 77.00 | 42.00 | 80.45 | 71.90 |
| | LoftQ | 4.00 | 16.00 | 15.33 | 69.35 | 53.92 | 78.83 | 84.25 | 75.10 | 78.17 | 43.00 | 77.37 | 64.17 |
| | LQ-LoRA | 3.85 | 16.00 | 18.52 | 66.91 | 53.70 | 81.70 | 81.25 | 44.00 | 77.80 | 43.80 | 80.50 | 72.55 |
| | AMQ+L | 3.91 | 16.00 | 14.90 | 70.86 | 53.10 | 81.40 | 82.10 | 74.80 | 77.90 | 43.40 | 81.10 | 73.10 |
| | AMQ+AL | 3.96 | 10.18 | 14.70 | 70.66 | 53.80 | 81.35 | 81.15 | 74.20 | 77.55 | 43.20 | 81.00 | 73.05 |
| | **AutoQRA ($\leq$4-bit)** | 3.71 | 10.57 | 11.95 | 71.35 | 54.30 | 81.26 | 83.74 | 74.86 | 78.46 | 43.60 | 81.30 | 73.30 |
| | **AutoQRA (Opt)** | 5.36 | 12.70 | 17.24 | **73.19** | 55.75 | 82.93 | 85.86 | 76.28 | 79.33 | 46.80 | 83.43 | 75.12 |
| Qwen-2.5-3B | LoRA | 16.00 | 16.00 | 10.84 | 65.53 | 48.20 | 75.10 | 80.00 | 64.00 | 75.50 | 35.00 | 77.00 | 69.40 |
| | QLoRA | 4.00 | 16.00 | 8.16 | 62.89 | 43.60 | 72.69 | 78.65 | 64.90 | 73.43 | 34.40 | 72.52 | 62.90 |
| | AdaLoRA | 4.00 | 15.92 | 8.70 | 62.51 | 42.50 | 75.57 | 76.09 | 68.30 | 70.70 | 28.20 | 74.62 | 64.10 |
| | LoftQ | 4.00 | 16.00 | 8.21 | 62.71 | 44.62 | 73.27 | 80.37 | 63.38 | 73.75 | 30.00 | 72.96 | 63.30 |
| | LQ-LoRA | 3.63 | 16.00 | 7.05 | 62.86 | **50.50** | 78.30 | 78.30 | 35.25 | 74.60 | **40.00** | 77.40 | 68.50 |
| | AMQ+L | 4.00 | 16.00 | 8.17 | 63.30 | 44.80 | 74.10 | 78.20 | 64.10 | 73.90 | 34.60 | 73.10 | 63.60 |
| | AMQ+AL | 3.80 | 15.84 | 8.65 | 64.88 | 45.60 | 76.54 | 77.90 | 72.50 | 71.80 | 31.40 | 76.13 | 67.20 |
| | **AutoQRA ($\leq$4-bit)** | 3.72 | 9.78 | 6.45 | 66.33 | 46.35 | 77.23 | 80.12 | 72.88 | 75.19 | 35.20 | 75.60 | 68.09 |
| | **AutoQRA (Opt)** | 5.22 | 12.00 | 8.31 | **68.05** | 47.31 | 79.32 | 82.21 | 74.00 | 77.52 | 36.80 | 76.86 | **70.40** |

setting, we evaluate $N_{\text{LF}} = 25$ configurations at the lowest fidelity $T_1$. Candidates are ranked for promotion using the surrogate screening model $\Phi_s$ (Eq. (10)) when enough paired data are available; otherwise we fall back to the measured $P(C; T_s)$. After successive halving, $N_{\text{HF}} = 3$ configurations are promoted to the highest-fidelity step count $T_S$ (high-fidelity, HF), *continuing training from their checkpoints* rather than restarting. All feasibility checks use the exact memory accounting in implementation.

### C.2. Phase I termination

We terminate Phase I using the hypervolume-based criterion in Eq. (28): if the relative improvement of dominated hypervolume stays below $\epsilon_{\text{hv}}$ for $\Delta$ consecutive generations, we stop and return the measured feasible non-dominated set $\mathcal{P}$.

After each generation, we update the population using NSGA-II with constrained domination: feasible candidates dominate infeasible ones; among infeasible candidates, lower constraint violation is preferred (Deb et al., 2002). To quantify progress of the feasible non-dominated set, we monitor the dominated hypervolume in the bi-objective space (maximize $P$, minimize $M$). We map each configuration to a minimization vector $\mathbf{f}(C) = \big( -P(C; T_S), M(C) \big)$ and define the hypervolume of a set $\mathcal{A}$ w.r.t. a reference point $\mathbf{r}$ as

$$\text{HV}(\mathcal{A}; \mathbf{r}) = \lambda \left( \bigcup_{\mathbf{a} \in \mathbf{f}(\mathcal{A})} [\mathbf{a}, \mathbf{r}] \right), \tag{27}$$
$$[\mathbf{a}, \mathbf{r}] = \{\mathbf{z} : a_k \leq z_k \leq r_k, \ \forall k\},$$

where $\lambda(\cdot)$ denotes the Lebesgue measure and $\mathbf{f}(\mathcal{A}) = \{\mathbf{f}(C) : C \in \mathcal{A}\}$. We use the relative hypervolume im-

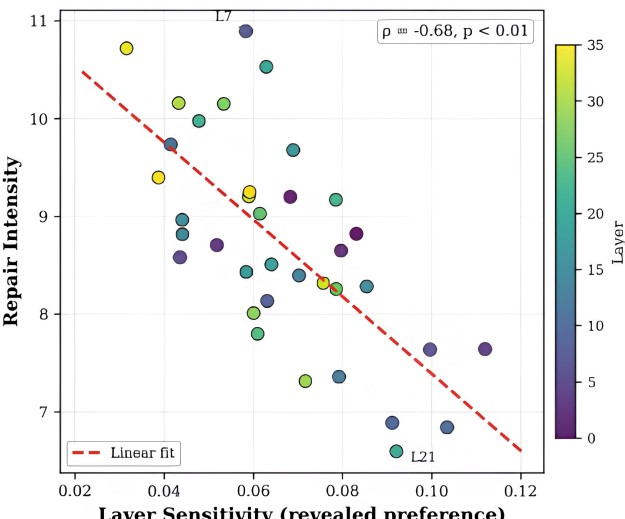

*Figure 8.* **Repair concentrates on robust layers.** Layer-wise repair intensity versus layer sensitivity (revealed preference). Each point is a layer (color indicates layer index); the dashed line is a linear fit. Repair intensity is strongly negatively correlated with sensitivity ($\rho = -0.68$, $p < 0.01$), showing that REPAIR$(\cdot)$ preferentially applies downgrades to less sensitive layers to satisfy $M(C) \leq B_{\max}$.

provement

$$\Delta\mathrm{HV}_g = \frac{\mathrm{HV}(\mathcal{P}_g; \mathbf{r}) - \mathrm{HV}(\mathcal{P}_{g-1}; \mathbf{r})}{\mathrm{HV}(\mathcal{P}_{g-1}; \mathbf{r}) + \epsilon_{\mathrm{hv}}^{\mathrm{den}}}, \qquad (28)$$

and stop Phase I if $\Delta\mathrm{HV}_g < \epsilon_{\mathrm{hv}}$ for $\Delta$ consecutive generations. Here $\epsilon_{\mathrm{hv}}^{\mathrm{den}}$ is a small constant to avoid numerical issues when the denominator is tiny. We return the final non-dominated feasible set $\mathcal{P}$ for Phase II.

### C.3. Phase II TuRBO high-fidelity evaluation

Phase II initializes $J$ trust regions from the $N_{\mathrm{HF}}$ measured points returned by Phase I. At iteration $t$, we form a discrete candidate pool $\Omega_t$ (Eqs. (14)–(15)) and compute $\mathrm{EI}_t(C)$ for all $C \in \Omega_t$ using the GP posterior. Only the maximizer $C_{t+1}$ in Eq. (17) is evaluated at the highest-fidelity step count $T_S$. We set a hard cap of $N_{\max}$ Phase II iterations. Optionally, we also cap the number of accepted HF evaluations per trust region by $N_{\max}^{\mathrm{TR}}$ (default $N_{\max}^{\mathrm{TR}} = 5$), so the worst-case additional HF evaluations in Phase II is at most $J \cdot N_{\max}^{\mathrm{TR}}$, but typically much smaller due to early stopping.

### C.4. Phase II early stopping

We stop Phase II when either (i) $\max_{C \in \Omega_t} \mathrm{EI}_t(C) < \epsilon_{\mathrm{ei}}$ (Eq. (20)), or (ii) the hard cap $N_{\max}$ is reached.

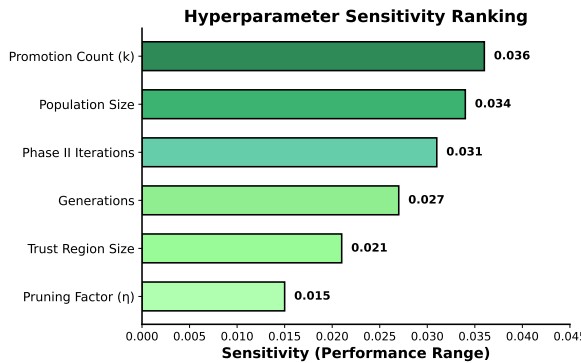

*Figure 9.* **Hyperparameter sensitivity ranking.** Sensitivity is measured as $S(h)$ in Eq. (29) (max–min performance across the sweep of $h$); larger values indicate higher sensitivity.

## D. Task-wise Accuracy Breakdown and Discussion

Table 4 breaks down accuracy by task for all four backbones. Averaged scores can hide task-specific trade-offs, so here we focus on the "shape" of the gains: which tasks stay stable under aggressive compression, and which ones are the first to fall apart.

**First takeaway: the uniform 4-bit baselines fail in very specific ways, and AutoQRA mostly fixes those failures.** Across backbones, QLoRA/AdaLoRA/LoftQ are often fairly close on the easier or more pattern-heavy tasks, but they can drop sharply on a few tasks that are more brittle to quantization noise. The cleanest example is **Qwen-2.5-7B**: QLoRA drops on **PIQA** (76.39) and especially **Wino-Grande** (63.30), even though its average remains competitive. AutoQRA ($\leq$4-bit) brings these back to 81.30 and 73.30, essentially recovering the FP16 behavior (81.66 / 73.04) while staying in the low-precision regime. So the main story is not that AutoQRA gives tiny uniform gains everywhere—it prevents the "one or two tasks drop sharply" pattern that you get with a rigid allocation.

**LLaMA-3.1-8B: AutoQRA ($\leq$4-bit) is close to "FP16-like" on most tasks, with the biggest win on GSM8K.** On this backbone, the uniform 4-bit methods lose most noticeably on **GSM8K** (e.g., QLoRA 44.15 vs. LoRA 54.06), while other tasks move less. AutoQRA ($\leq$4-bit) almost fully closes that gap (53.40), and it does so without trading away the rest: ARC-E (84.20 vs. 83.88), HellaSwag (79.50 vs. 79.44), and PIQA (82.40 vs. 82.10) are all on par with or slightly above FP16 LoRA. Interestingly, **OpenBookQA** is nearly flat across strong methods (AutoQRA ($\leq$4-bit) 45.60, LoRA 45.60), which suggests OBQA is not where the low-precision allocation hurts most for this model.

**LLaMA-3.2-3B: the gains are smaller and less uniform, but the "compensation" pattern still shows up where it**

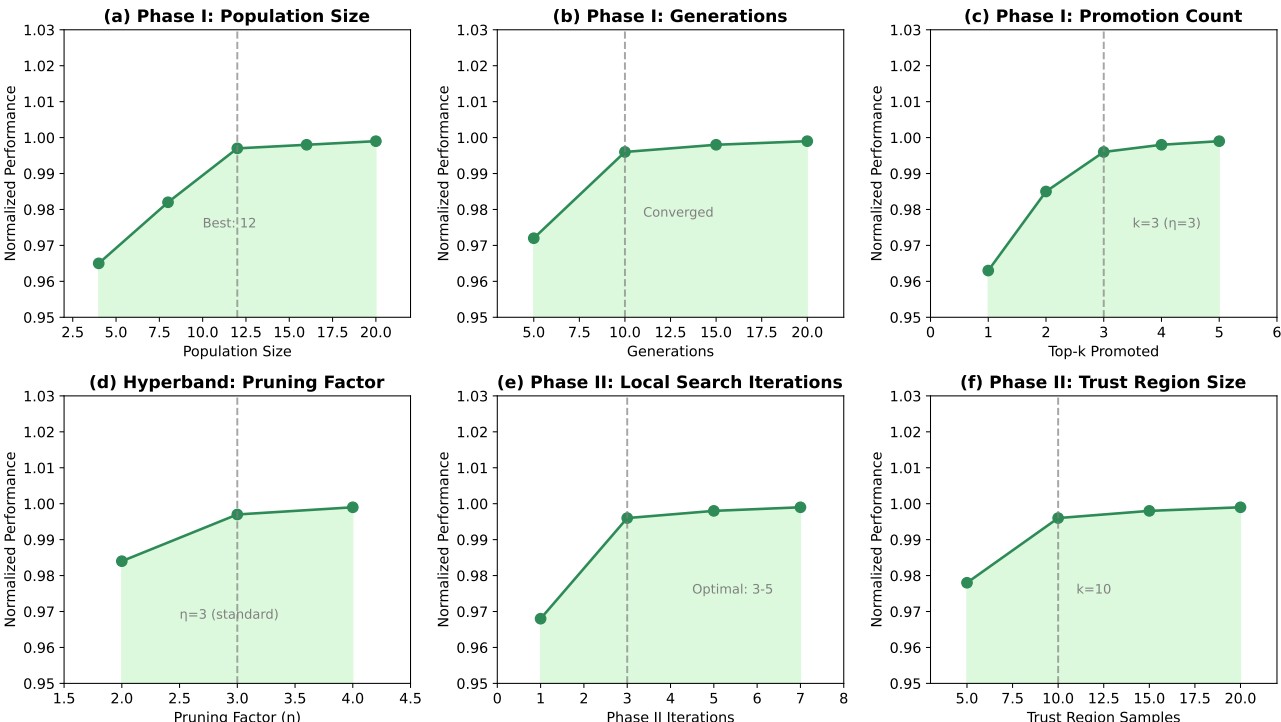

*Figure 10.* **One-factor hyperparameter sweeps.** Each panel varies one hyperparameter while keeping all others fixed. The dashed vertical line denotes the default setting used in all experiments; curves show mean performance.

**matters.** For the smaller LLaMA, AutoQRA ($\leq$4-bit) still improves **GSM8K** (48.93 vs. QLoRA 46.22 / LoRA 47.31), **HellaSwag** (76.97 vs. 74.79 / 75.75), and **WinoGrande** (71.20 vs. 70.96 / 70.09). At the same time, **BoolQ** drops for most low-precision methods and does not fully recover (AutoQRA (Opt) 79.70 vs. LoRA 81.10). This is a useful sanity check: once the backbone is small, joint allocation helps but cannot remove all capacity constraints. Some tasks remain sensitive to the exact capacity/precision trade, and the optimizer will sometimes accept a small loss on one task to gain more elsewhere (because the search objective is the post-fine-tuning average).

**Qwen-2.5-3B: AutoQRA reallocates capacity toward multi-step reasoning.** This backbone shows one of the clearest shifts on **GSM8K**: AutoQRA ($\leq$4-bit) reaches 72.88, substantially above LoRA (64.00) and well above uniform 4-bit baselines. At the same time, improvements on some other tasks are more modest (e.g., ARC-E 77.23 vs. 75.10, BoolQ 80.12 vs. 80.00), and a couple of tasks can move slightly in the opposite direction (e.g., PIQA 75.60 vs. 77.00). The overall picture matches the intended behavior of AutoQRA: it is not simply "quantize less everywhere," but rather it reallocates capacity so that the tasks that are most disrupted by quantization noise are the ones that get the most help from rank.

**AutoQRA ($\leq$4-bit) vs. AutoQRA (Opt): what the extra precision adds.** The (Opt) setting consistently raises the ceiling, but the pattern of gains matters. On **Qwen-2.5-7B**, the jump from AutoQRA ($\leq$4-bit) to (Opt) is broad (Avg 71.35 $\rightarrow$ 73.19), and it also improves the previously brittle tasks further (e.g., WinoGrande 73.30 $\rightarrow$ 75.12, PIQA 81.30 $\rightarrow$ 83.43). On **LLaMA-3.1-8B**, the (Opt) gains are more concentrated (ARC-C 56.12 $\rightarrow$ 57.85, ARC-E 84.20 $\rightarrow$ 84.90), while most other tasks are already near saturation under the $\leq$4-bit regime. So in practice, (Opt) mostly helps when you have headroom to improve harder tasks above an already-strong baseline, whereas $\leq$4-bit captures most of the robustness benefit under strict compression.

**Bottom line.** The per-task breakdown supports the core motivation: mixed bit-width choices that look acceptable in aggregate can still be fragile on specific tasks, and the joint bit–rank search largely removes these fragilities. Where the backbone has enough capacity (8B/7B), AutoQRA ($\leq$4-bit) approaches FP16 LoRA on most tasks while avoiding the sharp failure modes of uniform 4-bit methods. Where the backbone is small (3B), the trade-offs become more visible, but the improvements still show up consistently on the tasks that are typically hardest to preserve under quantization.

# E. Additional Ablations

## E.1. Feasibility Repair Analysis

Figure 8 gives a layer-level view of how REPAIR($\cdot$) behaves when it projects an infeasible configuration back into the feasible set. For each layer, we plot its sensitivity score (x-axis; a revealed-preference measure) against the corresponding *repair intensity* (y-axis; how strongly that layer tends to be hit by discrete downgrades during repair). The pattern is very clear: repair intensity is strongly *negatively* correlated with sensitivity ($\rho = -0.68$, $p < 0.01$). In other words, layers that are more sensitive are systematically protected, while downgrades are concentrated on layers that are relatively robust.

This is the behavior encouraged by Eq. (8). When $M(C) > B_{\max}$, REPAIR looks for the downgrade that removes the most memory for the least expected loss, i.e., small sensitivity-per-saved-memory. The scatter plot shows that this heuristic translates into a consistent, global preference rather than a few hand-picked cases: across depth (color-coded layer index), repair pressure shifts toward low-sensitivity layers, effectively using them as a buffer to satisfy the hard constraint without repeatedly harming fragile layers. A few outliers (e.g., the labeled layers) are expected in a discrete ladder setting: if a particular layer offers an unusually favorable memory drop for a single step on either the bit or rank ladder, it can be selected more/less often even when its sensitivity is not extreme. Overall, the takeaway is that REPAIR($\cdot$) enforces feasibility in a way that preserves trainability by avoiding heavy edits on sensitive layers.

## E.2. Search-Protocol Hyperparameter Sensitivity

AutoQRA contains several discrete hyperparameters that control (i) Phase I multi-fidelity evolutionary search (e.g., population size, number of generations, promotion rule, and pruning factor $\eta$), and (ii) Phase II local refinement (e.g., EI iteration budget, scalarization weight $\alpha$, and the neighborhood size in discrete trust regions). We assess robustness via a one-factor-at-a-time sweep: for each hyperparameter $h$, we vary $h$ over a candidate set $\Xi_h$ while fixing all other hyperparameters to their default values (marked by the dashed vertical line in the sweep plots). For each setting, we run the full AutoQRA pipeline under the same evaluation budget and report the resulting final performance (e.g., averaged task score; mean $\pm$ std over seeds).

To summarize sensitivity in a comparable scalar form, we define

$$S(h) \triangleq \max_{\xi \in \Xi_h} P^\star(\xi) - \min_{\xi \in \Xi_h} P^\star(\xi), \qquad (29)$$

where $P^\star(\xi)$ denotes the best measured final score returned by AutoQRA under setting $h = \xi$. Figure 9 ranks hyperparameters by $S(h)$, while Figure 10 shows the full response

*Table 5.* **Search and per-step overhead on Qwen2.5-3B.**

| Method | Search | Time/step (s) |
|---|---|---|
| QLoRA (4-bit) | 0 min | 2.41 |
| LoftQ (4-bit) | ~10 min | 2.42 |
| AdaLoRA (4-bit) | 0 min | 2.81 |
| AMQ+LoRA | >3.6 h | 1.91 |
| AutoQRA | 55 min | 1.92 |

*Table 6.* **Allocation granularity on Qwen2.5-3B.**

| Granularity | AvgBit | AvgRank | Search | Avg |
|---|---|---|---|---|
| Model-level | 4.00 | 16.00 | – | 62.89 |
| Block-level | 3.83 | 10.44 | 55 min | 64.85 |
| Layer-level | 3.72 | 9.78 | 55 min | 66.33 |
| Tensor-level | 3.68 | 9.52 | 3.6 h | 66.72 |

*Table 7.* **Hyperparameter tuning for Qwen2.5-3B.** Avg is computed over BoolQ, HellaSwag, PIQA, WinoGrande, and ARC-C.

| Setting | $\alpha$ | top-k | P2 iters | 5-task Avg | BoolQ | HellaS | PIQA | WinoG | ARC-C |
|---|---|---|---|---|---|---|---|---|---|
| Default | 0.5 | 3 | 3 | 69.07 | 80.12 | 75.19 | 75.60 | 68.09 | 46.35 |
| Tuned | 0.7 | 4 | 5 | 70.02 | 80.34 | 75.30 | 76.73 | 69.12 | 48.60 |

curves for each sweep.

**Takeaway.** Across the tested ranges, AutoQRA is generally stable around the default settings. We use the ranking (Fig. 9) to identify the most influential hyperparameters, and use the sweep curves (Fig. 10) to make the directionality of each effect explicit.

The sensitivity results support a simple tuning rule based on model capacity and task diversity. Smaller backbones, such as 3B models, are more constrained and benefit from a slightly wider promotion set and a few additional Phase II EI iterations. For larger backbones (7B and above), the default setting is usually sufficient because the model has more capacity to absorb quantization noise. When the target evaluation mixes heterogeneous tasks, increasing the promotion count and tuning the scalarization weight $\alpha$ can reduce task-level imbalance. For single-domain tasks, the default schedule is typically the most efficient choice.

# F. Extended Evaluation

**Search overhead.** Table 5 reports the wall-clock search cost and training time per step on Qwen2.5-3B. AutoQRA incurs an offline search cost, but it remains substantially lower than AMQ while jointly searching bit-width and rank. The final training loop has no dynamic rank-update overhead and is therefore close to AMQ+LoRA in per-step time.

**Granularity trade-off.** We compare model-, block-, layer-, and tensor-level allocation on Qwen2.5-3B in Table 6. Layer-level allocation provides the best cost–accuracy trade-off: tensor-level search gives only a small gain but expands the number of variables by a factor of seven and

*Table 8.* **Tighter-budget robustness on Qwen2.5-3B.** Avg is the combined average over BoolQ, HellaSwag, PIQA, WinoGrande, ARC-C, and HumanEval.

| Budget | Avg | BoolQ | HellaS | PIQA | WinoG | ARC-C | HumanEval |
|---|---|---|---|---|---|---|---|
| 1.0× | 64.55 | 80.12 | 75.19 | 75.60 | 68.09 | 46.35 | 41.96 |
| 0.7× | 63.92 | 79.48 | 75.23 | 75.52 | 67.25 | 44.63 | 41.39 |

*Table 9.* **Code-generation and 14B generalization.**

| Setting | Method | AvgBit | AvgRank | Metric |
|---|---|---|---|---|
| HumanEval | LoRA (FP16) | 16.00 | 16.00 | 43.19 |
| HumanEval | QLoRA (4-bit) | 4.00 | 16.00 | 40.90 |
| HumanEval | AMQ+LoRA | 4.50 | 16.00 | 24.40 |
| HumanEval | AutoQRA (≤4-bit) | 3.72 | 9.78 | 41.96 |
| HumanEval | AutoQRA (Opt) | 5.22 | 12.00 | 43.07 |
| Qwen2.5-14B | LoRA (FP16) | 16.00 | 16.00 | 80.31 |
| Qwen2.5-14B | QLoRA (4-bit) | 4.00 | 16.00 | 79.69 |
| Qwen2.5-14B | AMQ+LoRA | 3.88 | 16.00 | 79.08 |
| Qwen2.5-14B | AutoQRA (≤4-bit) | 3.75 | 10.50 | 80.30 |
| Qwen2.5-14B | AutoQRA (Opt) | 5.25 | 12.30 | 80.71 |

requires much longer search.

**Search budget on 3B models.** On Qwen2.5-3B, a wider promotion set and additional Phase II iterations improve the lower-scoring tasks while keeping the stronger metrics stable (Table 7).

**Tighter budgets and transferability.** Table 8 reports tighter-budget and transfer evaluations. When the memory budget is reduced to 0.7× the default ≤4-bit budget, performance degrades smoothly rather than collapsing. We also test whether a searched configuration can be reused across related datasets. Applying a configuration searched on Alpaca directly to HC3 obtains 65.54, compared with 65.93 for searching on HC3 itself and 62.45 for QLoRA, suggesting that part of the search cost can be amortized.

**Code generation and larger models.** Table 9 summarizes two generalization settings. On HumanEval after CodeAlpaca fine-tuning, AutoQRA matches FP16 LoRA with much lower average precision and rank. On Qwen2.5-14B, the ≤4-bit variant gives the smallest memory footprint while staying competitive, and the optimal variant reaches the best overall performance.

