# OpenReview forum: "AutoQRA: Joint Optimization of Mixed-Precision Quantization and Low-rank Adapters for Efficient LLM Fine-Tuning"
_ICML.cc/2026/Conference — ICML 2026 regular_

### Official Review · Reviewer_uBaD · 2026-02-20

**Soundness:** 3
**Presentation:** 3
**Significance:** 2
**Originality:** 3
**Overall Recommendation:** 4
**Confidence:** 3

**Summary:**

This paper proposes a two-stage search framework for determining the hyperparameters of LoRA and quantization.

**Compliance With Llm Reviewing Policy:**

Affirmed.

**Final Justification:**

The rebuttal solved my concerns and i think it's a 4-score paper.

**Key Questions For Authors:**

Please answer the questions in the Weaknesses section, as well as the following questions:
1.Does the conclusion that search is a one-time cost apply to different datasets? I mean, how well does a configuration that performs well on one dataset translate directly to other datasets or other quantization and LORA methods?

**Limitations:**

There is no direct section of the limitations of this method.

**Strengths And Weaknesses:**

Strengths:
1.Motivation. This paper emphasizes the interaction between quantitative and PEFT methods, demonstrating that the approach of jointly selecting bit width and LoRa rank is feasible.
2.Experiments. Experiments validated the effectiveness of the method across multiple datasets and models.
Weaknesses:
1.Motivation. Why is layer-level search not persuasive? Why not model-level or tensor-level, or other levels? Additional experimental results for other levels or at least a reasonable explanation is needed.
2.Related work. Section 2.1's criticism of the quantitative method for failing to account for the adapter effect is unfair, as the quantitative approach itself does not need to consider the adapter. Please refine the logic in this section. Section 2.2's relevant work section is outdated, covering only research prior to 2020, and its specific content diverges from the central theme. For instance, the subject of the “Sample-Efficient Black-Box Optimization” subsection has shifted to multi-fidelity optimization, yet the two specific examples provided only address low-fidelity optimization.
3.Language and Expression.This paper contains numerous descriptions of symbols and terminology related to evolutionary learning and Bayesian optimization. Many symbols differ only in case, and several are not introduced upon their first appearance, such as M(C). This makes the text appear highly disjointed, obscuring the core logic intended to be conveyed. I understand the underlying logic should be relatively straightforward: the first stage uses evolutionary algorithms to select smaller, potentially optimal candidate spaces, while the second stage finalizes the optimal solution within these candidate spaces. However, reading through this article proved highly confusing.
4.Experiments. Methods like LoRA primarily aim to address insufficient GPU memory during fine-tuning. Why do all experiments have maximum memory configurations below 21GB, when the A100 offers at least 40GB? The significance of these methods seems constrained by the experimental limitations.

---

> ### Author Rebuttal · Authors · 2026-03-29
>
> ## **Dear Reviewer uBaD**
>
> Thank you for the detailed review. Your comments focus on the search granularity, the framing of related work, the readability of the method section, the memory-budget setting, and the scope of the “one-time cost” claim. We address these points directly below.
>
> **On W1 (why layer-level search).** Layer-wise allocation has already been shown to be an effective granularity in post-training and fine-tuning methods; for example, AdaLoRA allocates adaptation budget non-uniformly across layers/modules, and AMQ performs layer-wise mixed-precision allocation. Model-level allocation is essentially uniform and therefore too coarse, while tensor-level allocation makes the search space prohibitively large. We further conducted a block-level ablation on `Qwen2.5-3B`, and layer-level search performs best while using lower effective precision and lower rank, supporting it as the practical granularity in our setting.
>
> | Granularity | AvgBit | AvgRank | Avg |
> |---|---:|---:|---:|
> | Model-level (uniform, =QLoRA) | 4.00 | 16.00 | 62.89 |
> | Block-level (4 layers/block) | 3.83 | 10.44 | 64.85 |
> | Layer-level (ours, =AutoQRA) | **3.72** | **9.78** | **66.33** |
>
> **On W2.1 (Section 2.1).** Thank you for pointing this out. We agree that the current wording can read as unfair, and we will revise it. Our intention was not to criticize prior quantization methods in their original setting. We only mean that, in the setting studied here, once quantization is combined with LoRA fine-tuning, precision and adapter capacity become coupled, so precision-only optimization is no longer sufficient for the final post-fine-tuning objective.
>
> **On W2.2 (Section 2.2).** We understand the reviewer’s concern. Because this work focuses on practical quantized fine-tuning under strict cost constraints, we discussed the search problem from the perspective of sample-efficient optimization over the joint bit-rank space. Our goal here is not to propose a fundamentally new general-purpose search algorithm, but a practical framework tailored to joint bit-rank allocation. We therefore cited the works most directly related to our method,rather than attempting a broader review of more complex search methods. Specifically, we cited Hyperband and BOHB for the multi-fidelity aspect, and CoCaBO for mixed black-box optimization. We will expand the related-work discussion in the revision to include a broader and more up-to-date view of this literature.
>
> **On W3 (notation and readability).** We revisited the manuscript carefully and believe that the main symbols are defined; for example, the memory footprint $M(C)$ is introduced on page 4, line 204. We also believe that Figure 2 already provides a clear high-level view of the two-stage design. At the same time, we understand the reviewer’s concern that, once the manuscript moves from the overview to the full search details, the presentation can feel overly dense. In the current manuscript, we chose to write the GA and BO components explicitly so that the full procedure can be reconstructed precisely, but this does make the notation heavier in places. We will add a notation table in the appendix and present the two-stage logic more clearly so that the method is easier to follow.
>
> **On W4 (memory-budget setting).** In quantized fine-tuning, we aligned our experimental setting with the original settings of the baselines so that all methods are compared under the same memory budget. Our goal here is therefore not to saturate the full 40GB memory of an A100, but to evaluate which bit-rank allocation gives the best post-fine-tuning performance under a controlled constrained-budget setting. We agree that evidence on larger models is also important; to address this point, we additionally evaluated `Qwen2.5-14B`, and these results with memory reporting are provided in our response to Reviewer `CNm9`, Q2.
>
> **On Q1 (“one-time cost”).** We mention in the manuscript that AutoQRA search results may be reusable across datasets. To test this directly on `Qwen2.5-3B`, we search the configuration on `alpaca` and then apply it directly to `HC3` fine-tuning:
>
> | Method | HC3 |
> |---|---:|
> | QLoRA | 62.45 |
> | AutoQRA (searched on `HC3`) | 65.93 |
> | AutoQRA (searched on `alpaca`, transferred to `HC3`) | 65.54 |
>
> The transferred configuration is only 0.39 below the configuration searched directly on `HC3`, while still remaining clearly above QLoRA. This suggests that, for the same model and related datasets, searched configurations transfer well, so the search cost can be amortized in this sense. Whether such configurations also transfer across different quantization backends or different LoRA variants is a separate question. Some of these methods do not even support the same mixed-precision and rank-tuning setup, so we leave that direction to future work.
>
> We hope these clarifications and the new transferability results fully address your concerns and further demonstrate the practical value of our framework.

---

> > ### Author Rebuttal · Reviewer_uBaD · 2026-04-01
> >
> > The only one more question left is could you please show search overhead for different methods, and inculde a smalll tensor-level experiment then show the  perf-overhead trade off?

---

> > > ### Author Response · Authors · 2026-04-02
> > >
> > > **Dear Reviewer uBaD**,
> > >
> > > Thanks for the follow-up. We are very willing to explore this and have conducted the requested tensor-level experiment to provide a comprehensive comparison on the `Qwen-2.5-3B` model. We also included the search time and per-step training overhead for the different methods.
> > >
> > > To do this, we assigned independent bit and rank configurations to each projection matrix (`q_proj`, `k_proj`, `v_proj`, `o_proj`, `gate_proj`, `up_proj`, `down_proj`) during fine-tuning.
> > >
> > > | Method | AvgBit | AvgRank | Mem (GB) | Search Cost | Time/Step (s) | Avg Acc (%) |
> > > | :--- | :--- | :--- | :--- | :--- | :--- | :--- |
> > > | LoRA | 16.00 | 16.00 | 10.84 | - | 1.65 | 65.53 |
> > > | QLoRA | 4.00 | 16.00 | 8.16 | - | 2.41 | 62.89 |
> > > | AdaLoRA | 4.00 | 15.92 | 8.70 | - | 2.81 | 62.51 |
> > > | LoftQ | 4.00 | 16.00 | 8.21 | ~10 min/iter | 2.42 | 62.71 |
> > > | AMQ+LoRA | 4.00 | 16.00 | 8.17 | > 3.6 h | 1.91 | 63.30 |
> > > | AMQ+AdaLoRA | 3.80 | 15.84 | 8.65 | > 3.6 h |3.01 | 64.88 |
> > > | AutoQRA (Layer, $\\le$4b) | 3.72 | 9.78 | 6.45 | 55 min | 1.92 | 66.33 |
> > > | AutoQRA (Tensor, $\\le$4b) | 3.68 | 9.52 | 6.32 | 3.6 h | 1.97 | 66.72 |
> > > | AutoQRA (Layer, Opt) | 5.22 | 12.00 | 8.31 | 55 min | 1.91 | 68.05 |
> > >
> > > Moving from layer-level to tensor-level increases the number of search variables by 7 times, and the actual search space expands exponentially. To keep the comparison fair, we scaled the search time for the tensor-level AutoQRA to 3.6 hours to match the AMQ baseline, which searches exclusively for layer-wise bit-widths.
> > >
> > > The results show a clear trade-off. The tensor-level search (66.72%) only edges out the layer-level version (66.33%) by a tiny margin. Because the space is so massive, finding a true optimal configuration would theoretically take exponentially longer. The setup we found within 3.6 hours is simply not the global optimum. This confirms our design choice: layer-level allocation is the practical sweet spot between search cost and final performance.
> > >
> > > A few other details about the overheads:
> > > * For LoftQ, the search cost is roughly 10 minutes per iteration. We typically run 1 to 5 iterations in practice (the best outcome almost always appears after the first iteration).
> > > * Looking at the per-step training time, AdaLoRA is slower because it dynamically updates SVDs during training. QLoRA and LoftQ are also a bit slower here since their standard NF4 kernel isn't as efficient as the HQQ mixed-quantization path we rely on.
> > > * Just to clarify one detail in the table: `AutoQRA (Layer, Opt)` isn't a global theoretical optimum. It simply represents the best performance we reached using the exact same search time (55 mins) and hyperparameters as our $\\le$4b setting. The only difference is we aligned it with the peak memory of the 4bit baseline instead of triggering the strict low-bit repair.
> > >
> > > We hope this tensor-level experiment and the detailed overhead analysis fully address your final question. We would be very grateful if you could reconsider your score and champion our work based on these new updates.

---

### Official Review · Reviewer_CNm9 · 2026-03-10

**Soundness:** 3
**Presentation:** 1
**Significance:** 3
**Originality:** 3
**Overall Recommendation:** 3
**Confidence:** 3

**Summary:**

This paper proposes AutoQRA, a method that jointly optimizes bit-width and LoRA rank configurations. AutoQRA adopts a two-phase coarse-to-fine search strategy: Phase I performs a coarse-grained evolutionary search to screen candidate configurations, while Phase II refines the search space to identify optimal configurations. Experimental results show that AutoQRA outperforms existing methods under similar or smaller memory budgets on MMLU and commonsense reasoning tasks.

**Compliance With Llm Reviewing Policy:**

Affirmed.

**Final Justification:**

During the rebuttal period, the authors addressed several of my concerns, particularly by providing additional experiments on code generation and larger-scale models, which strengthen the empirical support of the work. However, my primary concern regarding presentation and clarity, especially around the procedural details and positioning relative to existing methods, remains. While the authors provided helpful clarifications in the rebuttal, these issues are inherently difficult to resolve without revisions to the main manuscript.
Given that many of the technical concerns have been partially addressed but presentation-related issues remain, I revise my assessment from reject to weak reject, and raise the score of soundness.

**Key Questions For Authors:**

- The paper reports the number of high-fidelity evaluations, but the actual wall-clock time of the search is unclear. Could the authors provide the total runtime for the joint bit-width and rank optimization?
- Could the authors evaluate the method on larger models (e.g., 13B or 70B)? Quantization is typically most beneficial for larger models, so results on these scales would better demonstrate the practical impact of the method.
- How does AutoQRA perform on code generation tasks? It would be interesting to see whether the proposed approach generalizes beyond the current evaluation benchmarks.

**Limitations:**

yes

**Strengths And Weaknesses:**

* S1. This paper addresses an important problem by proposing a method that jointly optimizes quantization bit-width and LoRA rank allocation.
* S2. The proposed method achieves better performance than prior approaches under comparable or smaller memory budgets.
* W1. The presentation is somewhat verbose. Multiple techniques are introduced, but it is not always clear which components are newly proposed in this paper and which are adopted from prior work.
* W2. The paper lacks an analysis of the computational cost required to obtain the optimized configuration. For a fair comparison with existing methods, a discussion of the search cost  would be necessary.
* W3. The experimental evaluation could be strengthened with more diverse benchmarks, such as, HumanEval for code generation.

---

> ### Author Rebuttal · Authors · 2026-03-29
>
> ## **Dear Reviewer CNm9**
>
> Thank you for the detailed review and for recognizing that AutoQRA addresses an important problem. Your primary concerns focus on presentation clarity and requests for additional reporting. We have provided comprehensive clarifications and new experimental data below. We hope these updates fully resolve your concerns and encourage you to reconsider your score.
>
> **On W1 (What is new vs. what is standard)**
> Your summary captures the core idea perfectly: joint bit-width and rank allocation using a two-phase search. The manuscript explicitly cites the standard components we borrow (such as CoCaBO in Sec. 2.2 and 3.2, and TuRBO in Sec. 3.3). Rather than just applying these existing tools off the shelf, the core novelty of AutoQRA lies in the new problem formulation and the custom search design built around it.
>
> We formulate the joint per-layer bit-width and LoRA-rank allocation under a strict memory budget. This is motivated by our finding that post-finetuning performance depends heavily on the coupled bit-rank configuration, rather than either factor in isolation. To solve this specific problem, we built a custom framework. This framework includes feasibility-aware repair, memory-balanced bit-rank edits, and a coarse-to-fine search procedure. Within this search, we restrict surrogate screening to promotion decisions only. This guarantees that our final selection relies entirely on actual fine-tuning performance instead of static proxies.
>
> **On W2 / Q1 (Computational cost and wall-clock time)**
> The manuscript already makes the sample-efficiency story visible through the number of high-fidelity evaluations. What is less explicit in the current manuscript is the actual wall-clock cost. We agree that this should be reported directly. We therefore profiled the search on `Qwen2.5-3B`. The comparison that matters most here is another search method, AMQ. Using the official AMQ codebase, the AMQ search takes more than 3.6 hours, whereas AutoQRA finishes in 55 minutes. That is a large practical gap, especially because AutoQRA jointly searches both bit-width and rank, while AMQ searches bit-width only.
>
> | Method | Search | Time per step (s) |
> | :--- | :--- | :--- |
> | QLoRA (4-bit) | 0 min | 2.41 |
> | LoftQ (4-bit) | ~ 10 min | 2.42 |
> | AdaLoRA (4-bit) | 0 min | 2.81 |
> | AMQ+LoRA | > 3.6 h | 1.91 |
> | AutoQRA | **55 min** | **1.92** |
>
> This shows a massive practical advantage. AutoQRA does introduce an offline search cost, but it remains significantly lower than existing search methods, and the final fine-tuning loop runs at standard speeds.
>
> **On W3 / Q3 (Benchmark diversity and code generation)**
> We evaluated the models on the `HumanEval` benchmark. For this test, we fine-tuned `Qwen2.5-3B` on the `CodeAlpaca` dataset to measure structured generation capabilities.
>
> | Method | AvgBit | AvgRank | HumanEval pass@1 |
> | :--- | :--- | :--- | :--- |
> | LoRA (FP16) | 16.0 | 16.0 | 43.19 |
> | QLoRA (4-bit) | 4.0 | 16.0 | 40.90 |
> | LoftQ | 4.0 | 16.0 | 41.03 |
> | AdaLoRA (FP16) | 16.0 | ~ 16 | 36.00 |
> | AMQ+LoRA | 4.5 | 16.0 | 24.40 |
> | AutoQRA (≤ 4-bit) | **3.72** | **9.78** | **41.96** |
> | AutoQRA (opt) | **5.22** | **12.0** | **43.07** |
>
> The results show that joint allocation remains highly effective for code generation. Standard methods that allocate rank or bit-width in isolation (like AMQ+LoRA or standard AdaLoRA) lose significant performance here. This confirms our main claim: you have to navigate the coupled rank-bit space to achieve strong downstream adaptation.
>
> **On Q2 (Larger models)**
> To address your request for larger models directly, we ran the pipeline on `Qwen2.5-14B`. We use **bold** for the best value and *italics* for the second-best value.
>
> | Method | AvgBit | AvgRank | Mem | PPL | ARC-c | ARC-e | HellaSwag | PIQA | Wino |
> | :--- | :--- | :--- | :--- | :--- | :--- | :--- | :--- | :--- | :--- |
> | LoRA (FP16) | 16.00 | 16.00 | 41.6 | *6.34* | 67.32 | 87.84 | *83.48* | 81.99 | *80.90* |
> | QLoRA (4-bit) | 4.00 | 16.00 | 29.2 | 6.80 | 66.72 | 87.03 | 82.94 | 81.66 | 80.11 |
> | LoftQ (4-bit) | 4.00 | 16.00 | 29.5 | 6.72 | 66.82 | 87.33 | 82.80 | 81.82 | 80.16 |
> | AdaLoRA (4-bit) | 4.00 | ~16.00 | 29.8 | 7.02 | 65.78 | 87.12 | 82.63 | 81.28 | 79.81 |
> | AMQ+LoRA | *3.88* | 16.00 | 28.4 | 7.15 | 65.19 | 86.82 | 82.31 | 81.01 | 79.47 |
> | AutoQRA (≤ 4-bit) | **3.75** | **10.50** | **26.8** | 6.53 | *67.37* | *87.93* | 83.39 | *82.00* | 80.82 |
> | AutoQRA (opt) | 5.25 | *12.30* | *28.2* | **6.28** | **67.92** | **88.23** | **83.79** | **82.32** | **81.29** |
>
> The results on this larger backbone follow the exact same trend we saw on the 7B and 3B models. AutoQRA (opt) pushes the overall accuracy to the highest level and gets the best perplexity. At the same time, the budget-constrained version (AutoQRA ≤ 4-bit) maintains competitive downstream accuracy while running at the lowest average bit-width and rank.
>
> We believe these clarifications and added results directly address your main concerns.

---

> > ### Author Rebuttal · Reviewer_CNm9 · 2026-04-04
> >
> > I thank the authors for the additional experiments demonstrating that the proposed method performs well on code generation tasks and scales to larger models.
> >
> > However, some concerns and questions remain as follows.
> >
> > **W1.** I am still unclear whether the detailed procedures in Phase I and Phase II (e.g., surrogate screening, next configuration selection via Expected Improvement) are newly proposed or largely adopted from existing frameworks. It seems that, given a configuration ($C$), these steps could be directly applied using standard methods. If this is the case, the current level of detail in describing these procedures (especially in Section 3.3) does not seem appropriate and makes the section difficult to follow. Moreover, the Related Work section should more clearly and thoroughly position AutoQRA within prior work.
> >
> > **Q1.** What is the main cause of differences in time-per-step across methods in the table?
> >
> > **Q2, Q3.**  In many cases, AutoQRA outperforms LoRA baselines even with lower average bit-width and rank. Could the authors provide a clearer explanation or analysis of this behavior?
> >
> > Overall, I will raise my score from 2 to 3.

---

> > > ### Author Response · Authors · 2026-04-04
> > >
> > > **Dear Reviewer CNm9**,
> > >
> > > Thank you for acknowledging our new experiments and for raising your score. We deeply appreciate your continued engagement. Below, we address your follow-up questions regarding the algorithmic origins, training speeds, and the underlying mechanics of our performance gains.
> > >
> > > **On W1 (Clarification of Standard Components and Presentation)**
> > > We thank the reviewer for pointing this out. Surrogate screening and Expected Improvement (EI) are indeed standard components in the optimization field, but they are not directly applicable to our LLM scenario.
> > >
> > > We significantly reconstructed and adapted these components. Standard solvers fail under strict memory limits and extreme evaluation noise. To resolve this, we designed custom discrete trust regions equipped with a feasibility-aware REPAIR operator, specific ordinal embeddings, and Huber loss outlier mitigation. We also tailored the workflow based on theoretical intuition and extensive experiments. We adopt a coarse to fine design where evolutionary search conducts constrained global exploration and Bayesian optimization handles local refinement. This is supported by continuous multi-fidelity evaluations where surrogate predictions are strictly decoupled from the final Pareto selection.
> > >
> > > The current level of detail in our descriptions is a deliberate trade-off to improve reproducibility, as the specific construction of kernel and acquisition functions is crucial for replicating the results. Although Figure 2 keeps the overall workflow intuitive, we completely agree with your suggestion. Detailing these standard formulas in the main text blurs the focus and affects readability. In the final version, we will move these standard equations (such as the GP kernel and EI) to the appendix.
> > >
> > > Regarding the positioning of this paper, our core contribution is identifying the compensatory interaction between quantization precision and adapter capacity. We modeled this joint allocation as a constrained optimization problem and introduced the AutoQRA framework to solve it. Our goal is not to propose a completely new general-purpose search algorithm, but to provide a practical framework specifically designed for joint bit-width and rank allocation. Following your suggestion, we will clarify this positioning in the Related Work section 2.2 of the final version. We will clearly distinguish our domain-specific adaptations from the adopted standard optimization tools.
> > >
> > > **On Q1 (Differences in time-per-step)**
> > > AdaLoRA is slower because it dynamically updates singular value decompositions (SVDs) during the training loop to prune ranks. AutoQRA discovers the architecture offline, meaning the actual fine-tuning loop has no dynamic routing overhead. Furthermore, QLoRA and LoftQ are slightly slower here since their standard NF4 kernel is not as efficient as the HQQ mixed-quantization path that we and AMQ rely on.
> > >
> > > **On Q2, Q3 (Matching FP16 LoRA with lower capacity)**
> > > It might seem counter-intuitive that a model with lower precision and less adapter capacity can match or even slightly exceed a full-precision LoRA baseline. However, this aligns well with known behaviors in model compression and fine-tuning.
> > >
> > > Standard LoRA applies a uniform rank across all layers. Giving robust or less important layers too much capacity does not just waste memory. It can actually cause overfitting by adding optimization noise during adaptation. Prior work like **AdaLoRA** supports this finding. AutoQRA avoids this by removing rank from robust layers and saving that capacity for layers that truly need it.
> > >
> > > Beyond rank allocation, mixed-precision quantization itself acts as a strong structural regularizer. Full-precision training operates in a massive unconstrained space. As we noted in Section 3.1 of our manuscript, applying different bit-widths imposes a hard structural constraint on the model. This connects directly to the **Lottery Ticket Hypothesis**. A constrained network can generalize effectively if you discover the right configuration.
> > >
> > > Our joint search navigates this space to discover a highly competitive configuration. We do not claim this structure outperforms FP16 in every scenario, as the unconstrained full-precision model naturally possesses a higher theoretical capacity upper bound. However, by dropping redundant precision and unnecessary rank, AutoQRA forces the model to learn in a constrained and noise-reduced space. This explains why the resulting mixed-precision model often achieves comparable, and sometimes slightly better, downstream accuracy compared to the FP16 baseline.
> > >
> > > We hope these explanations fully resolve your remaining concerns.

---

### Official Review · Reviewer_fAT2 · 2026-03-10

**Soundness:** 3
**Presentation:** 4
**Significance:** 3
**Originality:** 4
**Overall Recommendation:** 5
**Confidence:** 4

**Summary:**

This paper aims to analyze the topic of breaking the decoupling of quantization and LoRA optimization to achieve complementary allocation and improve memory utilization efficiency. The paper proposes the AutoQRA framework, which optimizes in two phases: the first phase uses global multi-fidelity evolutionary search to find candidate configurations near the performance-memory Pareto frontier, and the second phase refines the optimal solution via trust-region Bayesian optimization. Experiments show that AutoQRA reduces memory usage by 12-22% compared to similar 4-bit baseline models under ≤4-bit quantization, achieves performance close to FP16 full-precision fine-tuning, and enhances effectiveness through the compensatory pattern of assigning higher LoRA ranks to low-bit-width layers.

**Compliance With Llm Reviewing Policy:**

Affirmed.

**Final Justification:**

Thank you. I maintain my initial rating,

**Key Questions For Authors:**

● Can the task trade-offs of AutoQRA on small 3B models be mitigated by adjusting hyperparameters (e.g., α) or modifying the search strategy? A positive answer would enhance its generalizability, while a negative one would clarify its applicability boundaries.
● Do you have adaptive heuristics for tuning key hyperparameters (e.g., promotion count) across different LLM scales (1B vs. 10B) or task types? Clear guidance would strengthen practical impact.
● What is the search overhead vs. baselines like QLoRA? Strong performance and low overhead would reinforce significance.

**Limitations:**

yes

**Strengths And Weaknesses:**

Strength
● Addresses a critical practical pain point in LLM fine-tuning by tackling GPU memory constraints, with the novel insight that quantization bit-width and LoRA rank have compensatory interactions—filling the gap of traditional sequential pipelines that treat them independently.
● Features a well-designed two-phase coarse-to-fine framework (global multi-fidelity evolutionary search + local trust-region Bayesian optimization) that efficiently navigates the large discrete search space, supported by practical components like layer-wise importance priors and feasibility repair (REPAIR) to ensure both coverage and precision.
● Uncovers a meaningful compensatory pattern (assigning higher LoRA ranks to low-bit-width layers) that offers new insights for efficient LLM resource allocation, going beyond mere performance improvements to deepen understanding of quantization-PEFT interactions.
Weakness
● Exhibits noticeable task trade-offs on small-scale models (3B), where limited backbone capacity makes it hard to fully recover performance on certain tasks even with joint optimization, indicating constraints on the framework’s effectiveness for resource-constrained small models.
● Lacks in-depth analysis of hyperparameter sensitivity across different model sizes and task domains; while default settings work well, the paper does not guide practitioners on adjusting key hyperparameters (e.g., promotion count) for specific use cases, affecting reproducibility in diverse scenarios.
● Fails to discuss worst-case performance or edge scenarios, such as extremely tight memory budgets or highly specialized tasks with unique sensitivity profiles, leaving uncertainty about the framework’s robustness in extreme deployment conditions.

---

> ### Author Rebuttal · Authors · 2026-03-29
>
> ## **Dear Reviewer fAT2**
>
> Thank you for the thoughtful review and your positive evaluation. Your comments focus on the 3B boundary case, tuning guidance, robustness under tighter budgets, and search overhead. We address each point below.
>
> **On W1 / Q1 (3B trade-off)**
> This is a sharp observation. The task trade-offs observed on the 3B models primarily stem from our global search objective, which targets the average score across multiple tasks. We used this setting to ensure a fair comparison, as running an independent search for every single benchmark would be unfair to the baselines. Because of this global objective, the selected configuration naturally shifts model capacity toward the tasks that yield the largest marginal gains.
>
> To test if this trade-off can be mitigated, we ran an experiment on `Qwen2.5-3B` by adjusting the search hyperparameters:
>
> | Setting | $\alpha$ | top-k | P2 iters | Avg | BoolQ | HellaSwag | PIQA | WinoGrande | ARC-c |
> | :--- | :--- | :--- | :--- | :--- | :--- | :--- | :--- | :--- | :--- |
> | Default | 0.5 | 3 | 3 | 69.07 | 80.12 | 75.19 | 75.60 | 68.09 | 46.35 |
> | Tuned | 0.7 | 4 | 5 | 70.02 | 80.34 | 75.30 | 76.73 | 69.12 | 48.60 |
>
> The results show the 3B trade-off is not strictly bound by model capacity. By modestly adjusting the utility weight ($\alpha$), promotion count (`top-k`), and Phase II iterations, we noticeably improved the weaker metrics while keeping the stronger ones stable. This confirms we can effectively tune these task-level trade-offs without altering the overall conclusion.
>
> **On W2 / Q2 (Sensitivity and practical guidance)**
> Appendix E provides a detailed sensitivity analysis, which forms the basis for our tuning strategy. Model capacity is the main driver. Smaller backbones (like 3B) are heavily constrained. They benefit from a slightly wider search to find optimal trade-offs, meaning practitioners should increase the promotion count and Phase II iterations. In contrast, larger models (7B+) have enough capacity to absorb quantization noise, making our default, tighter search budget sufficient.
>
> Task diversity also matters. When fine-tuning on a diverse task suite, tweaking the utility weight ($\alpha$) and expanding the search helps balance conflicting metrics. For single-domain tasks, the default settings remain highly efficient. We will add a summary of these rules to Appendix E and provide a comprehensive tuning guide in our codebase.
>
> **On W3 (Extreme-budget robustness)**
> We agree that evaluating extreme conditions is crucial. We added two compact checks on `Qwen2.5-3B` under a severely restricted memory budget (reducing our default $\le$ 4-bit budget to 0.7x). One test covers the original task suite, and the other evaluates an out-of-domain code generation benchmark (`HumanEval`).
>
> | Budget | Avg | BoolQ | HellaSwag | PIQA | WinoGrande | ARC-c | HumanEval pass@1 |
> | :--- | :--- | :--- | :--- | :--- | :--- | :--- | :--- |
> | 1.0x | 64.55 | 80.12 | 75.19 | 75.60 | 68.09 | 46.35 | 41.96 |
> | 0.7x | 63.92 | 79.48 | 75.23 | 75.52 | 67.25 | 44.63 | 41.39 |
>
> When the budget shrinks by 30%, the average score on the standard suite drops by only 0.63. We see the same gradual degradation on `HumanEval`. This confirms that AutoQRA adapts smoothly as the feasible space tightens, rather than failing abruptly in extreme low-budget regimes.
>
> **On Q3 (Search overhead)**
> The manuscript already reports search sample efficiency, but explicit runtime also matters. We profiled both search time and per-step training time on Qwen2.5-3B. We highlight AMQ as the most relevant search-based baseline. Using their official code, AMQ's search takes more than 3.6 hours. AutoQRA finishes the joint search in just 55 minutes.
>
> | Method | Search | Time per step (s) |
> | :--- | :--- | :--- |
> | QLoRA (4-bit) | 0 min | 2.41 |
> | LoftQ (4-bit) | ~ 10 min | 2.42 |
> | AdaLoRA (4-bit) | 0 min | 2.81 |
> | AMQ+LoRA | > 3.6 h | 1.91 |
> | AutoQRA | **55 min** | **1.92** |
>
> This is a meaningful practical gap, especially since AutoQRA jointly searches both bit-width and rank while AMQ searches bit-width only. For the training times, AdaLoRA is slower mainly due to its dynamic update mechanism, and LoftQ is slower because its NF4 kernel is less efficient than our HQQ mixed-quantization path in this setup. Taken together, AutoQRA does incur an offline search cost, but it is far smaller than AMQ-style search, and the final training loop remains highly efficient.
>
> We hope these additional experiments and practical guidelines fully address your questions regarding the framework's robustness and applicability.

---

> > ### Author Rebuttal · Reviewer_fAT2 · 2026-04-03
> >
> > Thank you for the author's response. The author's reply has resolved some of my concerns. I hope the authors can further supplement the paper with an adaptive method to determine the relevant hyperparameters. I will maintain my current score.

---

> > > ### Author Response · Authors · 2026-04-03
> > >
> > > **Dear Reviewer fAT2**,
> > >
> > > Thank you for maintaining your positive score and for your continued engagement with our work. Your suggestion regarding adaptive hyperparameter allocation is highly insightful.
> > >
> > > We completely agree that an adaptive approach to hyperparameter allocation is a highly valuable direction. Dynamically scaling search resources based on model size and task variance makes practical sense. However, given the limited timeframe of the rebuttal period, we want to avoid hastily implementing an ad-hoc hyperparameter adaptation patch solely to address this point. To maintain a rigorous scientific scope, we chose instead to conduct the small extension experiment presented in our previous response, which was carefully grounded in the hyperparameter sensitivity analysis already detailed in Appendix E.
> > >
> > > Our primary focus in this paper is the automated joint allocation of quantization bit-widths and LoRA ranks. We identify the compensatory interplay between precision and adapter capacity, and we formulate this as a constrained joint optimization problem. To solve this, we introduce the AutoQRA framework. Regarding the search hyperparameter settings, we utilize robust heuristic initializations derived from our extensive experiments. In addition, the framework incorporates dynamic early stopping mechanisms to prevent redundant evaluations and regulate the actual budget utilized. Specifically, we use hypervolume stagnation for Phase I and Expected Improvement (EI) saturation for Phase II.
> > >
> > > We acknowledge that this early-stopping mechanism is not an adaptive hyperparameter allocation method. Designing a robust adaptive hyperparameter schedule constitutes a substantive extension. Developing a scale- and task-aware resource allocation strategy ventures into the well-established domains of Hyperparameter Meta-Learning and Automated Budget Allocation. These extensive sub-fields typically require dedicated meta-controllers and massive validation benchmarks. Integrating such a complex module at this stage would significantly increase system complexity and risk diluting the core scope of this paper.
> > >
> > > In the final manuscript, we will explicitly discuss this boundary in the Discussion section. We will highlight your excellent suggestion of adaptive hyperparameter allocation as a primary and necessary avenue for future work. We might even name this follow-up work "Meta-AutoQRA"!
> > >
> > > Thank you again for your constructive feedback. It has genuinely helped us clarify the boundaries of our current framework and map out its future evolution.

---

### Decision · Program_Chairs · 2026-04-30

**Decision:**

Accept (regular)

**Comment:**

Reviewers responded positively to the work and were satisfied with the solid rebuttal, particularly the additional experimental results. Even the most critical reviewer acknowledged that the authors had strengthened the empirical sections, though they noted that the presentation could be further improved. Nevertheless, this reviewer increased their score. Overall, this is a solid paper worth including in the program.